# Using a long short-term memory (LSTM) neural network to boost river streamflow forecasts over the western United States

Kieran M. R. Hunt[1,2], Gwyneth R. Matthews[1,3], Florian Pappenberger[3], and Christel Prudhomme[3,4,5]

[1]Department of Meteorology, University of Reading, UK
[2]National Centre for Atmospheric Sciences, University of Reading, UK
[3]European Centre for Medium-Range Weather Forecasts, Reading, UK
[4]Department of Geography and Environment, Loughborough University, UK
[5]UK Centre for Ecology and Hydrology, Wallingford, UK

**Correspondence:** Kieran M. R. Hunt (k.m.r.hunt@reading.ac.uk)

**Abstract.** Accurate river streamflow forecasts are a vital tool in the fields of water security, flood preparation and agriculture, as well as in industry more generally. Traditional physics-based models used to produce streamflow forecasts have become increasingly sophisticated, with forecasts improving accordingly. However, the development of such models is often bound by two soft limits: empiricism – many physical relationships are represented empirical formulae; and data sparsity – long time-series of observational data are often required for the calibration of these models.

Artificial neural networks have previously been shown to be highly effective at simulating nonlinear systems where knowledge of the underlying physical relationships is incomplete. However, they also suffer from issues related to data sparsity. Recently, hybrid forecasting systems, which combine the traditional physics-based approach with statistical forecasting techniques, have been investigated for use in hydrological applications. In this study, we test the efficacy of a type of neural network, the long-short term memory (LSTM), at predicting streamflow at ten river gauge stations across various climatic regions of the western United States. The LSTM is trained on the catchment-mean meteorological and hydrological variables from the ERA5 and GloFAS-ERA5 reanalyses as well as historical streamflow observations. The performance of these hybrid forecasts is evaluated and compared to the performance of both raw and bias-corrected output from the Copernicus Emergency Management Service (CEMS) physics-based Global Flood Awareness System (GloFAS).

Two periods are considered, a testing phase (June 2019 to June 2020), during which the models were fed with ERA5 data to investigate how well they simulated streamflow at the ten stations; and an operational phase (September 2020 to October 2021), during which the models were fed forecast variables from ECMWF's Integrated Forecasting System (IFS), to investigate how well they could predict streamflow at lead times of up to ten days.

Implications and potential improvements to this work are discussed. In summary, this is the first time an LSTM has been used in a hybrid system to create a medium-range streamflow forecast, and in beating established physics-based models, shows promise for the future of neural networks in hydrological forecasting.

# 1 Introduction

Accurate forecasts of river streamflow are vital across a range of sectors, including, but not limited to, agriculture, water security, recreation, disaster management, and heavy industry. As such, modelling streamflow as a function of observable hydrological and meteorological variables has been the subject of focused study for nearly 200 years, and has intensified considerably over the past few decades as demands on water resources continue to increase dramatically (Beven, 2011).

The earliest attempt (Mulvaney, 1851) comprised a simple linear relationship between streamflow and catchment rainfall, derived using linear regression. Key early developments then split the catchment into regions based on estimated travel time to the gauge (Imbeaux, 1892; Ross, 1921) and included more variables in the regression model (Linsley et al., 1949). One of the first physics-based streamflow models was developed by Horton (1933), who considered the role of excess soil filtration in runoff. Since then, physics-based models have largely dominated, particularly with continued improvements to process understanding (e.g. Freeze and Harlan, 1969), computing power (e.g. Kollet et al., 2010; Schiemann et al., 2018), observation systems – including discharge data vital for model calibration (e.g. Newman et al., 2015), and remote sensing (e.g. Huffman et al., 1995; Robock et al., 2000).

Although physics-based models have improved substantially they are still currently limited by lack of process understanding, lack of – particularly subsurface – data, and inadequate grid resolution (Wood et al., 2011). Such problems can be overcome by the application of artificial neural networks, which can produce highly accurate simulations of physical systems even if the underlying physical relationships are not known. In hydrology, a particular type of artificial neural network, known as a Long Short-Term Memory network (LSTM; Hochreiter and Schmidhuber, 1997; Gers et al., 2000), has become increasingly popular due to its ability to process sequential data (Shen and Lawson, 2021).

LSTMs are a special case of so-called recurrent neural networks (RNNs), i.e., artificial neural networks that are capable of processing data containing temporal sequences. The basic feature of RNNs is a feedback loop that allows the network to retain information over time. However, due to their relatively simple construction, RNNs do not readily retain long-term temporal dependencies (Hochreiter and Schmidhuber, 1997). This is overcome in LSTMs by the addition of a special unit, known as a memory cell or the forget gate, that can preserve information indefinitely, allowing LSTMs to learn long-term dependencies that other RNNs cannot. In modelling river discharge over the United States, Kratzert et al. (2018) showed that LSTMs vastly outperform conventional RNNs.

For the reasons outlined above, it is unsurprising that LSTMs are being increasingly used to explore complicated hydrological problems including modelling river streamflow. In this regard, studies fall into two categories – either seeking to create a model capable of replicating existing streamflow observations, or seeking to create a model capable of forecasting streamflow at some future time. This distinction is minor in a machine learning context, since replication is a form of prediction; however, they have important distinctions in their application to the hydrological community. Several highly illustrative studies approach the former topic. In particular, a series of papers published by researchers at Johannes Kepler University Linz (Kratzert et al., 2018, 2019a, b; Klotz et al., 2021; Gauch et al., 2021b) demonstrated the remarkable ability of LSTMs to simulate daily streamflow in catchments across the United States and in the quantification of streamflow uncertainty. Extending this work,

Gauch et al. (2021a) showed that reanalysis-trained LSTMs could predict streamflow at any given temporal resolution, and at multiple resolutions simultaneously. These studies demonstrated that simple LSTM architectures (i.e. unstacked) performed as well as stacked ones in simulating streamflow. They also showed that better performance could be achieved by training a single LSTM over many basins and including data on basin geography than by individual LSTMs trained on a per-basin basis, even extending to good performance in ungauged basins.

The latter topic has proven more challenging, with only a handful of studies trying (to the authors' knowledge) to use LSTMs to predict streamflow (Slater et al., 2021). The most basic of these rely on rivers where streamflow has a strong annual cycle and large lagged autocorrelation (i.e. high persistence), using only antecedent streamflow data from the same site. Such studies have mixed, although promising results (de Melo et al., 2019; Sahoo et al., 2019; Sudriani et al., 2019; Zhu et al., 2020). More advanced models also incorporate upstream data, either just streamflow at different sites (Silva et al., 2021), or additionally precipitation (Le et al., 2019; Hu et al., 2020) with improved results. Le et al. (2019), for a case study in Vietnam, and Silva et al. (2021), for a case study in Brazil, achieved good results at 3- and 5-day lead times respectively. Ding et al. (2019) produced perhaps the most sophisticated LSTM-based hydrological forecast model to date. A LSTM that ingested ECMWF forecasts of precipitation, soil moisture, and other variables to produce a runoff forecast around the confluence region of the Lech and Danube Rivers. They verified forecasts up to lead times of nine hours, finding a Nash-Sutcliffe efficiency of 0.71, rising to 0.77 after the inclusion of an attention mechanism. The use of the ECMWF forecasts (created using a physics-based numerical weather prediction system) to feed an LSTM (a machine learning method) puts the Ding et al. (2019) study into the realm of hybrid forecasting.

The definition of a hybrid forecasting system is broad and currently blurry in the literature, partly due to the cross-disciplinary nature of the topic and partly due to the wide range of opportunities for integrating the two approaches into a single forecasting system (e.g., see Düben et al., 2021). Generally speaking, in hydrology, a hybrid hydrological forecasting system is one which incorporates physically based and statistical or machine learning methods. Combining the contrasting approaches is anticipated to provide superior forecasts by compensating for the limitations of each approach when used independently (some of which are discussed above). However, this definition is subjective and depends on where you draw the boundaries around the hydrological forecasting system. Is the numerical weather prediction considered part of the hydrological forecasting system or as input data (e.g., Ding et al., 2019; Liang et al., 2018)? If the model calibration of a physical model is aided by machine learning is this process included in the definition of a hybrid system (e.g., Teweldebrhan et al., 2020)? Would deep-learning-based quality control of observations be included within a hybrid-system (e.g., Sha et al., 2021)? Other potential inclusions into the class of hybrid forecasting systems are machine-learning aided data assimilation (Boucher et al., 2019; He et al., 2020; Liu et al., 2021), and commonly post-processing (e.g., Liu et al., 2022; Sharma et al., 2021; Lee and Ahn, 2021; Frame et al., 2021; Nearing et al., 2020). Answering these questions is beyond the scope of this study and requires a discipline-wide collaboration to definitively define a hydrological forecasting system. Therefore, we restrict our discussions to the less disputed part of the hydrological forecasting chain: the representation of the runoff-routing processes.

The hybridisation of hydrological models within the forecasting chain can be achieved in multiple ways. The simplest approach would be to combine the output from two hydrological models, one physically based, and one machine learning

based (Wagena et al., 2020; Booker and Woods, 2014). Alternatively, machine-learning based and process based models can be coupled. This can either be sequentially (e.g., Noori and Kalin, 2016) or in parallel with sub-models representing different processes (Reichstein et al., 2019; Bennett and Nijssen, 2021; Meng et al., 2016; Okkan et al., 2021).

More recently, more integrated methods of hybridisation have been introduced to hydrological modelling. Theory-guided (or physics-informed) machine learning approaches (Karpatne et al., 2017; Raissi et al., 2019) aim to overcome the interpretability issues identified with 'black-box' machine learning methods and constrain the output to theoretically plausible outputs (Xu et al., 2021; Chadalawada et al., 2020). Hoedt et al. (2021) included mass-conservation as a constraint into an LSTM architecture and found that for streamflow, although a decrease in the Nash-Sutcliffe Efficiency metric was seen, the mass-conserved LSTM better predicted extreme flow values (e.g., flood peaks). However, Frame et al. (2022) found that the addition of the mass-constraint reduced the skill when predicting extreme values suggesting further work is necessary. Alternatively, other studies have used neural networks in process-based models to solve differential equations more efficiently whilst allowing interpretability of the output (Höge et al., 2022; Rackauckas et al., 2020; Raissi et al., 2020).

In this study, we seek to expand on these previous efforts and develop an LSTM capable of providing skilful river streamflow forecasts at lead times of up to ten days at ten stations across the western United States. We will train the model mainly using meteorological variables from the ERA5 reanalysis (Sec. 3.1) and hydrological variables from GloFAS-ERA5 (Sec. 3.3, meaning that we are not at the mercy of potentially sparse observational data, but will use official gauge observations (Sec. 2) as the target to give optimal calibration. Once trained, the LSTM will be used to produce forecasts by replacing ERA5 inputs with forecast variables from the ECMWF Integrated Forecast System (IFS; Sec. 3.2). Again the use of the IFS means that the LSTM forecasts are not vulnerable to data latency of observations in an operational setting. Additionally, since the IFS is used to drive the ERA5 reanalysis, any significant climatological biases in one are likely to be present in the other. Since we are training the model with ERA5 data, such biases – so long as they are consistent between the two products – will be mitigated as the LSTM either applies an internal bias correction, or gives the field a low weighting in the input layer. We train the model on publicly-available ERA5, rather than IFS hindcasts, so that our methods can be completely reproduced by any interested reader. The forecasts will be made under operational time and data constraints for a thirteen month period (September 2020 to October 2021) and the results compared with GloFAS (Sec. 3.3), a physics-based streamflow forecast produced by ECMWF; a new, bias-corrected version of GloFAS (Sec. 4.3); and a simple persistence model. The core aims of this study are to determine (a) whether such an LSTM based hybrid system can provide skilful streamflow forecasts, (b) whether a hybrid system can perform better than existing state-of-the-art physics-based systems, and (c) whether advanced bias-correction techniques can improve the skill of physics-based models. An important caveat here is that seven of the ten catchments are smaller than the recommended usage threshold for GloFAS (2000 km$^2$), and so we would probably expect to see improvement when adding degrees of freedom through bias-correction or hybridising with an LSTM.

The study is laid out as follows: we discuss the study region and the climatological characteristics of the ten gauge stations in Sec. 2. We then describe the data used in Sec. 3 and methods – including the bias-correction algorithm and the LSTM setup – in Sec. 4. The results section is split into two parts, verification of a testing phase – where the models are driven with ERA5

– in Sec. 5.1, and verification of the operational phase – where the models are driven with IFS output – in Sec. 5.2. Finally, we discuss potential applications and improvements to our work in Sec. 6 and conclude with a summary in Sec. 7.

## 2   Study region and choice of stations

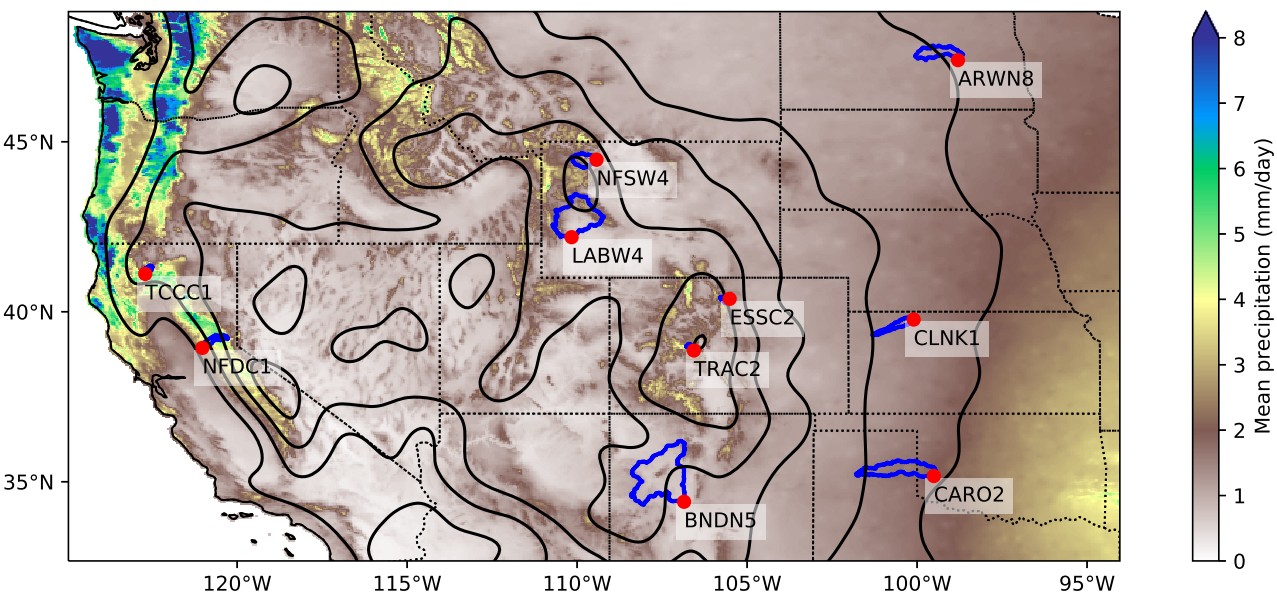

**Figure 1.** Locations of the ten streamflow gauges (red) and their catchment basins (dark blue). Overlaid are the climatological precipitation (filled contours, 1981–2010, from PRISM), smoothed orography (black line contours at 500 m intervals) from ETOPO, and state boundaries (dotted black lines).

The choice of study region and stations was dictated by the US Department of Reclamation, part of whose remit is water security over the western half of the contiguous United States. Between September 2020 and September 2021, they sponsored

a competition (https://www.topcoder.com/community/streamflow) to predict streamflow at ten locations (Fig. 1 and Tab. 1) at lead times of up to ten days, into which we entered three forecasts, raw and bias-corrected GloFAS and an LSTM, which are discussed in greater detail in Sec. 4, and which form the basis of this study.

The ten gauge locations are shown in the context of climatological precipitation (from PRISM; Daly et al., 2008) and topography (from ETOPO1 Amante and Eakins, 2009) in Fig. 1, along with their respective catchment basins in blue. The ten

locations represent a considerable diversity in climate and environment: two stations (TCCC1, NFDC1) are on the windward side of the Sierra Nevada, a region of high climatological rainfall; four (NFSW4, LABW4, ESSC2, TRAC2) are situated at high elevations along the Rockies; and four (ARWN8, CLNK1, CARO2, BNDN5) are in the relatively arid plains of the Midwest and south. These locations are summarised in Tab. 1.

| Location ID | USGS Station Number | Description | Longitude | Latitude |
|---|---|---|---|---|
| BNDN5 | 8353000 | Rio Puerco near Bernardo, NM | -106.85 | 34.41 |
| ARWN8 | 6468250 | James River above Arrowwood Lake near Kensal, ND | -98.80 | 47.40 |
| TCCC1 | 11523200 | Trinity River above Coffee Creek near Trinity Center, CA | -122.70 | 41.11 |
| CARO2 | 7301500 | North Fork Red River near Carter, OK | -99.51 | 35.17 |
| ESSC2 | 6733000 | Big Thompson River above Lake Estes, CO | -105.51 | 40.38 |
| NFDC1 | 11427000 | North Fork American River at North Fork Dam, CA | -121.02 | 38.94 |
| LABW4 | 9209400 | Green River near La Barge, WY | -110.16 | 42.19 |
| CLNK1 | 6847900 | Prairie Dog Creek above Keith Sebilius Lake, KS | -100.10 | 39.77 |
| TRAC2 | 9107000 | Taylor River above Taylor Park, CO | -106.57 | 38.86 |
| NFSW4 | 6279940 | North Fork Shoshone River at Wapiti, WY | -109.43 | 44.47 |

**Table 1.** Summary of the locations of the ten river streamflow gauges used in this study. See also Fig. 1.

| Station ID | Basin area (km$^2$) | Elevation (m) | Discharge (m$^3$ s$^{-1}$) | | | | Missing (%) |
|---|---|---|---|---|---|---|---|
| | | | 25% | median | mean | 75% | |
| BNDN5 | 13739 | 1439 | 0.000 | 0.000 | 0.368 | 0.042 | 23.8 |
| ARWN8 | 1165 | 439 | 0.195 | 1.44 | 8.24 | 9.34 | 50.6 |
| TCCC1 | 386 | 773 | 12.1 | 17.2 | 20.5 | 24.1 | 1.6 |
| CARO2 | 5367 | 508 | 0.821 | 1.95 | 3.28 | 3.82 | 9.0 |
| ESSC2 | 357 | 2290 | 0.680 | 1.76 | 4.47 | 5.30 | 22.2 |
| NFDC1 | 885 | 217 | 24.2 | 37.4 | 49.0 | 53.2 | 4.1 |
| LABW4 | 10123 | 1987 | 18.3 | 21.2 | 26.0 | 28.6 | 37.1 |
| CLNK1 | 1527 | 712 | 0.059 | 0.122 | 0.142 | 0.187 | 14.3 |
| TRAC2 | 331 | 2847 | 0.963 | 1.05 | 1.19 | 1.39 | 33.3 |
| NFSW4 | 1810 | 1700 | 4.76 | 6.17 | 8.55 | 10.1 | 18.5 |

**Table 2.** Summary of the climatological hydrological parameters of the ten river streamflow gauges used in this study. 'Missing' indicates the fraction of measurements since 01 Jan 1990 recorded either as 'Ice' or some other non-numeric value. Data from USGS and DWR, as outlined in Sec. 3.4.

Selected hydrological statistics are shown for each gauge in Tab. 2. These are computed over the entire measurement record
for each gauge (minimum ∼30 years) and reflect the variance in river characteristics chosen by the Bureau of Reclamation. Notably, only three stations (BNDN5, CARO2, LABW4) have a drainage basin whose area exceeds 2000 km$^2$; this is the recommended threshold for GloFAS analysis, as basins smaller than this are not necessarily resolved by the underlying model.

| Station ID | Mean 2-m temp (°C) | | | Mean precip (mm d$^{-1}$) | | |
|---|---|---|---|---|---|---|
| | Annual | Jan | Jul | Annual | Jan | Jul |
| BNDN5 | 10.9 | -1.6 | 22.7 | 1.0 | 0.8 | 1.8 |
| ARWN8 | 4.9 | -12.7 | 20.6 | 2.0 | 0.8 | 3.3 |
| TCCC1 | 8.0 | -0.4 | 20.2 | 3.4 | 6.6 | 0.5 |
| CARO2 | 15.6 | 3.5 | 27.6 | 1.8 | 0.9 | 1.6 |
| ESSC2 | 1.9 | -8.4 | 14.8 | 2.1 | 1.3 | 3.1 |
| NFDC1 | 10.2 | 2.2 | 21.4 | 4.3 | 8.6 | 0.1 |
| LABW4 | 1.6 | -9.9 | 16.4 | 1.5 | 1.7 | 0.7 |
| CLNK1 | 12.0 | -1.0 | 25.8 | 1.9 | 0.6 | 2.8 |
| TRAC2 | -0.9 | -11.4 | 11.7 | 1.7 | 1.7 | 1.7 |
| NFSW4 | 0.0 | -10.6 | 14.2 | 2.0 | 2.0 | 1.2 |

**Table 3.** Summary of the climatological meteorological parameters of the ten river streamflow gauges used in the competition. Data from ERA5, as outlined in Sec. 3.1.

The remaining seven stations, therefore, provide us with a forecast challenge. As Tab. 2 shows, some gauges are in extremely arid locations (e.g. BNDN5), and some have a great quantity of missing data (e.g. ARWN8), both of which present potential difficulties in the training and operational use of the LSTM. Overall (Tab. 3), the basin-average meteorology varies considerably between the gauges, reflecting their geographical diversity.

## 3 Data

### 3.1 ERA5

The Copernicus Climate Change Service (C3S) at ECMWF produces the five-generation ERA5 atmospheric reanalyses of global climate covering the period since January 1950 (Hersbach et al., 2020). Data from ERA5 (available from https://apps. ecmwf.int/mars-catalogue/?class=ea) cover the entire globe on a 30 km grid and resolve the atmosphere on 137 levels from the ground up to 80 km in altitude. At reduced spatial and temporal resolutions, ERA5 includes uncertainty information for all variables. We use catchment-mean ERA5 variables (near-surface, surface, and subsurface) as training data for the LSTM.

### 3.2 IFS

The study uses the ECMWF Integrated Forecasting System (IFS, version CY47R1). This IFS was run at full complexity with the configuration used for operational weather forecasts at ECMWF, as well as for the re-analysis (ERA5). The system is described in detail (https://www.ecmwf.int/en/publications/ifs-documentation) and has a turbulent diffusion and exchange with the surface represented by the Monin-Obukhov similarity theory in the surface layer and an Eddy-Diffusivity Mass-Flux

(EDMF) framework above the surface layer and includes a mass-flux shallow-convection; a multilayer, multitiled land-surface scheme (HTESSEL); a five-species cloud microphysics model; and a shortwave and longwave radiation scheme including cloud radiation interactions. IFS data, up to a 10-day lead time, are used as input to the LSTM when it is run operationally (available from https://apps.ecmwf.int/mars-catalogue/?class=od&stream=oper). Each morning throughout the operational period (September 2020 to October 2021), the control member (the member generated with unperturbed initial conditions) of the ensemble was downloaded using the Meteorological Archival and Retrieval System (MARS) API. This comprises more than 20 variables, globally, at a six-hourly frequency and resolution of $0.1 \times 0.1°$. Having to download and pre-process this large volume of data within the time constraints allowed limited us to using only one ensemble member. To retain more realistic variability at longer lead times, we chose to use the control member, rather than the ensemble mean. The full list of variables used is given in Sec. 4.4.

### 3.3 GloFAS

The worldwide Global Flood Awareness System (GloFAS; Harrigan et al., 2020), created collaboratively by the European Commission and the European Centre for Medium-Range Weather Forecasts (ECMWF), is a global hydrological forecasting and monitoring system that is not constrained by administrative or political boundaries. It combines cutting-edge meteorological predictions with a hydrological model, and due to its continental scale setup, it can deliver information on upstream river conditions as well as continental and global overviews to downstream nations. Since 2011, GloFAS has been producing daily ensemble flood predictions and monthly seasonal streamflow outlooks since November 2017 and is run operationally as a component of the Copernicus Emergency Management Service. For dates prior to May 25 2021 (i.e. all of the testing period and most of the operational period), we use GloFAS version 2.1 (Zsoter et al., 2019a), thereafter we use GloFAS version 3.1 (Zsoter et al., 2021). GloFAS products are freely available from its dedicated Information System, open to all following registration (www.globalfloods.eu) and its hydrological data from Copernicus Climate Data Store (https://cds.climate.copernicus.eu/#!/home). More detail on GloFAS service can be found on the dedicated wiki (https://confluence.ecmwf.int/display/CEMS/Global+Flood+Awareness+System). Two sets of GloFAS data were used: GloFAS-ERA5 (Zsoter et al., 2019b), a global modelled dataset of daily river discharge created by forcing the hydrological model with the ERA5 reanalysis; and GloFAS forecasts (Zsoter et al., 2019a, 2021), an ensemble of global daily river discharge forecasts, forced by ensemble forecasts from the ECMWF IFS. As with the IFS above, for forecasts we use the control member up to a lead time of ten days, downloaded using the MARS API (available from https://apps.ecmwf.int/mars-catalogue/?class=ce&stream=wfas). The control member of GloFAS is the ensemble member produced using the control member of the IFS ensemble forecast. These data are global, have daily frequency, and a resolution of $0.1 \times 0.1°$.

### 3.4 Observational station data

Observational gauge data were downloaded from https://dwr.state.co.us/Tools/Stations (for ESSC2) and https://waterdata.usgs.gov/nwis (all others). These data are available at three-hourly resolution, published in near real-time, and mostly with coverage

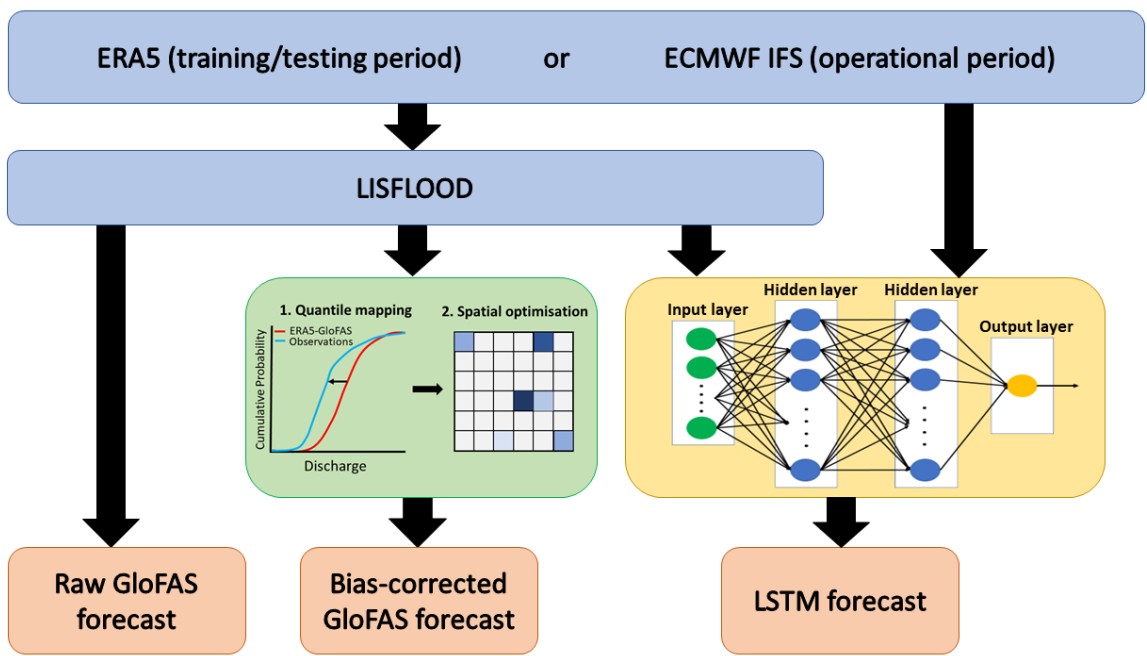

**Figure 2.** Schematic showing the workflow of the raw GloFAS forecasts, the bias-corrected GloFAS forecast, and the LSTM hybrid forecast.

from about 1990 onwards. Coverage and streamflow data are given in Tab. 2. These data are used to train the LSTM and calibrate the parameters for both stages of the bias-correction algorithm (see Sec. 4.3), and later for forecast verification.

## 4 Methods

In this section we describe the verification and forecasting method used in this study. Three forecasts are created each day with 6-hourly timesteps out to a maximum lead-time of 10 days. The overall workflow used to create these forecasts is shown in Fig. 2. More detail is provided in the following sections.

First, we describe the different time periods used in this study to train and validate the LSTM and bias-correction methods.

### 4.1 Training, testing, and operational forecasting periods

We define three periods which are used consistently throughout the study.

- The **training period** runs from January 1990 to June 2019. Although some stations have observational data prior to 1990, this start date was chosen to ensure a consistent length of training dataset across all gauges and to reduce the effect of spurious or sparse data on the training algorithms. The end date was chosen so that a twelve-month testing period was available before the start of the operational forecasting period. The same training period and dataset is used for the quantile mapping, spatial optimisation, and LSTMs.

– The **testing period** runs from June 2019 to June 2020, fixed as a twelve-month period that ended just before the opera-
tional phase was due to begin.

     – The **operational forecasting period** runs for thirteen months from September 2020 to October 2021. The start and end
date were fixed by the competition.

## 4.2   Verification metrics

The Kling-Gupta Efficiency (KGE, Gupta et al., 2009; Kling et al., 2012) is used to evaluate the performance of the forecasts.
The KGE is defined as

$$\text{KGE} = 1 - \sqrt{(r-1)^2 + (\beta-1)^2 + (\gamma-1)^2} \tag{1}$$

where $r$ is the Pearson's correlation coefficient, $\beta$ is the bias ratio, and $\gamma$ is the variability ratio. The bias and variability ratios
are defined as

$$\beta = \frac{\mu_{\text{sim}}}{\mu_{\text{obs}}}, \tag{2}$$

and

$$\gamma = \frac{\sigma_{\text{sim}}}{\sigma_{\text{obs}}}, \tag{3}$$

where $\mu_{\text{sim}}$ and $\sigma_{\text{sim}}$ are the mean and standard deviation of the forecast discharge and $\mu_{\text{obs}}$ and $\sigma_{\text{obs}}$ are the equivalent for
the observed discharge. The KGE is widely used in hydrology as each component is a measure of a different type of error. The
correlation coefficient, $r$, is a measure of temporal errors, the bias ratio, $\beta$ indicates whether the discharge tends to be over-
or under-predicted by the forecast, and the variability ratio, $\gamma$ measures if the forecast captures the variability of the discharge
magnitudes (Harrigan et al., 2020).

    For a perfect forecast, each component ($r$, $\beta$, and $\gamma$) has a value of 1, giving KGE= 1. We also define two benchmarks. We
define a 'skilful' forecast as one where the KGE is higher than for a mean observed discharge benchmark i.e., KGE$> 1-\sqrt{2} \sim$
$-0.414$. However, beating the long-term climatological mean is often easy (Pappenberger et al., 2015) and for certain flow
regimes not even a helpful metric (Knoben et al., 2019) – although in our case, failing to surpass this benchmark does clearly
indicate a lack of skill. To this end, we also arbitrarily define a 'highly skilful' forecast as one where KGE$> \sqrt{2}/2 \sim 0.707$.
This benchmark corresponds to a mean relative error in the three coefficients of about 17%, or an error of about 30% in one
coefficient if the other two have zero error.

As an additional upper benchmark, we will also use persistence forecasts. Persistence forecasts can help us to determine
which stations are 'easy' or 'hard' to forecast. For each station, these are constructed by persisting its mean observed discharge
from the previous 48 hours. For example, the 5-day persistence forecast for May 23 is given by the mean flow between May
16 and May 18. Evaluated at a fixed lead time, such persistence forecasts asymptotically approach the observed mean and
standard deviation of the observations over long periods, typically giving them high KGEs.

As such, we also use the Nash-Sutcliffe Efficiency Nash and Sutcliffe (NSE; 1970), which validates forecast or simulated flow based only on covariance with the observations, thus:

$$\text{NSE} = 1 - \frac{\Sigma_t (Q_{\text{sim}}(t) - Q_{\text{obs}}(t))^2}{\Sigma_t (Q_{\text{obs}}(t) - \overline{Q_{\text{obs}}(t)})^2}, \tag{4}$$

where $Q_{\text{sim}}$ is the simulated discharge, $Q_{\text{obs}}$ is the observed discharge, and the overbar denotes a long-term average.

## 4.3   Bias correction

When undertaking bias correction, we have a range of choices of complexity – ranging from the very simple (additive/multiplicative) through increasingly advanced methods (e.g. quantile mapping). Here, we have the advantage of a long timeseries of training data and we want to maximise the forecast skill under the single constraint that forecast output from GloFAS is the only input. To that end, we employ both quantile mapping and spatial fitting techniques, splitting the bias correction into two serial algorithms, which we outline in the following subsections. Quantile mapping remaps modelled values to reduce systematic

bias and is a standard practice in meteorological and hydrological bias correction (Thrasher et al., 2012). The spatial optimisation method, newly developed for this work, provides an additional layer of bias correction, accounting for the fact that consistent spatiotemporal biases in hydrometeorological fields such as precipitation (should they exist), will result in consistent upstream/downstream biases in modelled streamflow — information that can be used to improve forecasts.

### 4.3.1   Quantile mapping

For the first stage of the bias correction we employ a basic quantile mapping method. The training period (January 1990 to June 2019) was extracted from the observational record. GloFAS-ERA5 streamflow was extracted for the same period, not only for the grid point in which the gauge of interest is located, but also for surrounding points in a $0.6° \times 0.6°$ box centred on the gauge. This gives a total of 36 locations (given the $0.1°$ spacing of global GloFAS output).

    Iteratively, these are then quantile-mapped to the observed streamflow, that is:

$m_{\text{bc}}(i, t) = \tilde{q}_{\text{obs}}(q_{\text{raw}}(m_{\text{raw}}(i, t))), \tag{5}$

where $q$ is a function that maps streamflow to streamflow quantiles, $\tilde{q}$ is its inverse, $m_{\text{raw}}$ and $m_{\text{bc}}$ are the raw and bias-corrected modelled streamflows respectively, and $i$ and $t$ are spatiotemporal indices. The forms of $q_{\text{raw}}$ and $q_{\text{obs}}$ are both computed using data from the training period and used unchanged for the testing period and operational forecasts. For example, consider a forecast streamflow value of 38.7 m$^3$ s$^{-1}$ for a grid point containing the NFSW4 gauge. This value, were it in the GloFAS-

260 ERA5 training period, would have a quantile value of 0.88. The 0.88th quantile (or 88th percentile) for observed streamflow at NFSW4 in the training period is 77.0 m$^3$ s$^{-1}$, and so this would be the value used for $m_{\text{bc}}$ for that point at that forecast time. This makes sense, given that raw GloFAS underestimates high flow at NFSW4 by about 50% (cf. Fig. 5). This quantile mapping technique is then carried out independently for each of the 36 grid points in the neighbourhood of each gauge, in each case mapping the GloFAS output for the specific grid point to the observations taken at the gauge (as opposed to observations

taken at the grid point itself).

#### 4.3.2 Spatial optimisation

We must then convert these forecast values over the neighbourhood grid points into a single forecast value. This was achieved through a simple linear summation, i.e.,

$$m_{\mathrm{bc}}(t) = \Sigma_i a_i m_{\mathrm{bc}}(i,t) \,, \tag{6}$$

where the coefficients $a_i$ are to be determined.

To compute $a_i$, eq. 6 was treated as an optimisation problem using the same training period as earlier (January 1990 to June 2019). Here, we seek to minimise

$$-\mathrm{NSE}(m,o) - \mathrm{KGE}(m,o) \,, \tag{7}$$

i.e. the negative sum of the Kling-Gupta and Nash-Sutcliffe efficiencies over the training period. This optimisation was carried
out subject to the constraint that $0 \le a_i \le 1$, to prevent unphysical behaviour, and computational noise. We minimise this combination of NSE and KGE because using NSE alone leaves the optimization procedure vulnerable to incorrect local minima (e.g., an incorrect mean). However, we also found that using KGE alone tended to result in a bias correction that weighted correct mean and variance too highly compared to correlation, which is not useful for improving forecasts. We found that combining the two improved the weighting more in favour of correlation, while avoiding spurious local minima. Because the
number of training datapoints is several orders of magnitude higher than the number of fitting parameters, and because of the constraints we impose on $a_i$, overfitting is very unlikely. This is confirmed later by validation over the testing period in Sec. 5.1.2.

A sequential quadratic programming technique was then used to compute the optimal bias matrix, $a_i$, due to its ability to obey constraints through Lagrange multipliers (Nocedal and Wright, 2006). The resulting (static) $6 \times 6$ matrices were computed
and stored for each of the ten gauges and are then used during each forecast to convert the quantile-mapped $n_t \times 6 \times 6$ forecasts into 1D vectors of length $n_t$.

As an example, the bias matrix for NFSW4 is:

$$\begin{bmatrix} 0.089 & 0 & 0 & 0 & 0.138 & 0 \\ 0 & 0 & 0 & 0 & 0 & 0 \\ 0 & 0 & 0 & 0 & 0 & 0 \\ 0 & 0 & 0 & 0.754 & 0.009 & 0 \\ 0 & 0 & 0 & 0 & 0 & 0 \\ 0 & 0 & 0.001 & 0 & 0 & 0.023 \end{bmatrix}. \tag{8}$$

It is clear that the gauge is located in the grid point associated with $a_i = 0.754$; however, there are some additional contributions
from nearby river points. Aside from the direct spatial error described in the previous subsection, these uncentred contributions (both in NFSW4 and at other stations) largely work to correct two smaller biases. Firstly, any local spatial bias in the ERA5 precipitation used to drive GloFAS-ERA5 that, for example, incorrectly increases the streamflow in a nearby channel. Secondly,

a temporal bias in the streamflow, for example a downstream point may receive nonzero weighting if the modelled streamflow at the gauge is occurring later than in observations. This method has the additional advantage that no topological data is required.

Two final adjustments are made when the bias correction is run operationally. Firstly, the bias-corrected forecast is slightly relaxed towards the raw forecast ($q_{\text{bc}}^{\text{new}} = 0.25q_{\text{raw}} + 0.75q_{\text{bc}}^{\text{old}}$) to account for the different climatologies of GloFAS-ERA5 and GloFAS. GloFAS-ERA5 and GloFAS forecasts have different climatologies because they take meteorological input from different sources (ERA5 and IFS forecasts respectively). ERA5 and IFS themselves have different climatologies because, although they share the same driving model, ERA5 is a reanalysis and is nudged towards observations, whereas IFS forecasts are not. At the beginning of the operational phase, we noticed that the differences between GloFAS and GloFAS-ERA5 were leading to an overenthusiastic bias correction. The damping ratio was chosen by testing the effect of selected values in the range 0–1. Secondly, the whole forecast is then shifted by an additive $\delta$, the difference between the mean observations and the mean day-1 forecasts over the two days prior to the forecast being issued.

## 4.4 LSTM

### 4.4.1 Summary

For the reader's reference, we provide a brief summary of the LSTM and its use. The LSTM is trained separately on each of the ten basins in Tab. 1 using historical observations of streamflow and catchment-mean hydrological and meteorological variables from ERA5, giving 23 training variables (Tab. 5) and one target variable (streamflow) in total.

For each gauge, the LSTM is trained 100 times using random starting weights, with the mean of the five best-performing ensemble members from the training period used for validation during the testing period and forecasting during the forecast period. We differ from the suggested best practice of the JKU papers (see Introduction) of using a single LSTM trained over many basins only because the majority of our code base had been written and models trained by the time those papers were published.

During the testing period, the LSTMs are run at six-hourly timesteps, ingesting catchment-mean ERA5 variables from the preceding week at a six-hourly resolution (giving an input sequence of length 28, fixed during the training process). This process results in a single estimate of streamflow for each timestep at each gauge. During the forecast period, we want to provide daily streamflow forecasts out to a ten-day lead time at six-hourly resolution (i.e. 40 timesteps). To achieve this, at each gauge, the LSTM is run 40 times daily and the outputs concatenated to build the desired timeseries. For each timestep, the input sequence – still of length 28 – is constructed from catchment-mean IFS forecast variables. For example, a day-4 forecast to be issued on March 15, valid on March 19, would use an input sequence comprising data from the following IFS forecasts and lead times: March 13 day-0, March 14 day-0, March 15 day-0, March 15 day-1, March 15 day-2, March 15 day-3 and March 15 day-4.

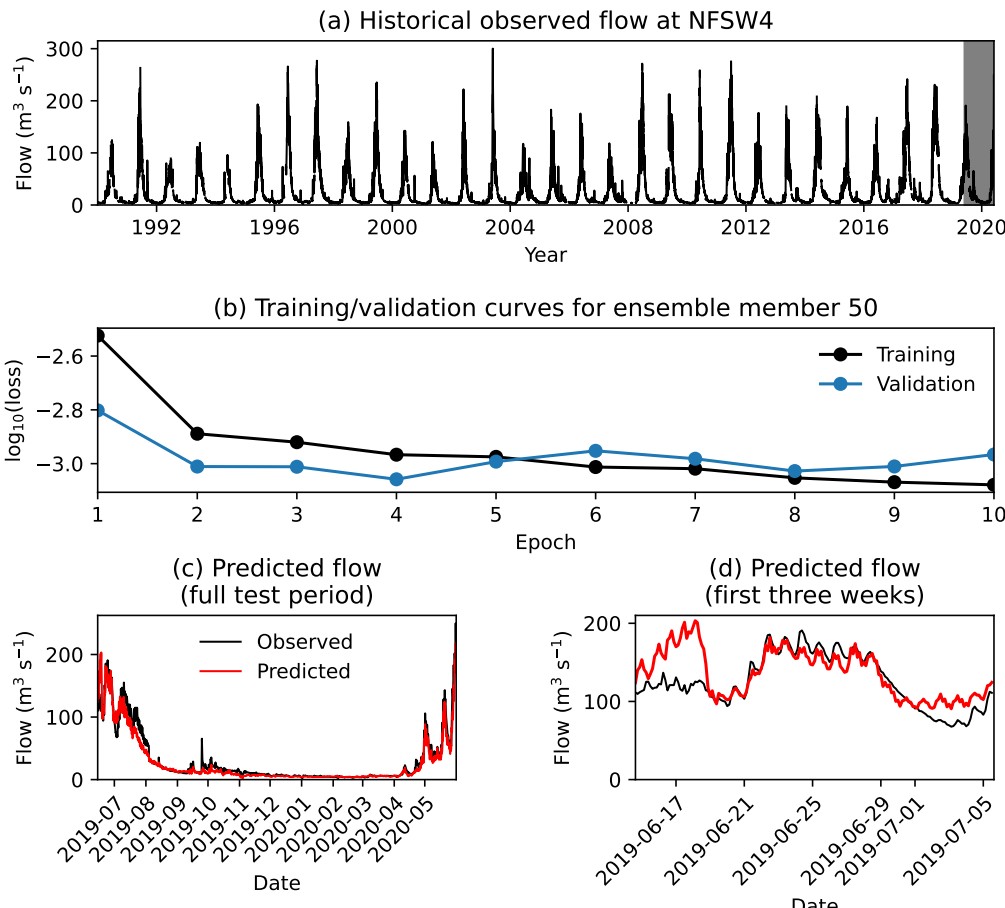

**Figure 3.** Example training of a single LSTM ensemble member for the NFSW4 gauge, showing (a) the full dataset of observed streamflow, with the testing period shaded in grey; and (b) comparison of the training and validation losses over the ten training epochs. For a fair comparison, the validation loss has been multiplied by the ratio of the variances of the testing and training datasets. Also shown are comparisons of the predicted streamflow to observations for the full (c) and start (d) of the testing period.

### 4.4.2 Setup and training

A number of hyperparameters are required for an LSTM and our selection was largely guided by previous literature (e.g. Kratzert et al., 2018). This included: using mean squared error as our loss function; a learning rate of 0.001 with no decay; and a dropout of 0.1. For a full inventory of hyperparameters, see the code repository link at the end of the paper. We decided to use basin-mean variables so that the LSTM architecture would be consistent for each of the ten gauges. With this setup fixed, we tested stacked architectures with permutations of 5, 10, 20, 50, and 100 neurons in each LSTM layer. The best performance, judged using the testing period at each gauge, was achieved by the $50{\times}50{\times}50$ permutation, giving the architecture described in Tab. 4.

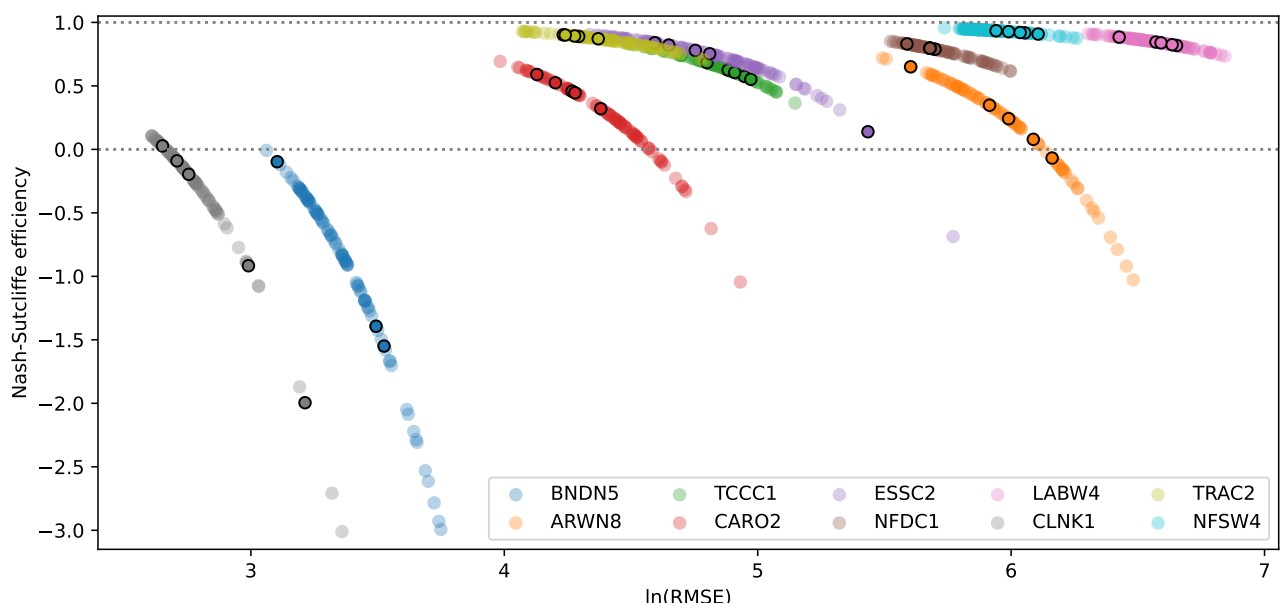

**Figure 4.** The Nash-Sutcliffe efficiency and logarithm of the root mean square error in streamflow (m$^3$ s$^{-1}$) for each of the hundred ensemble members of each gauge-specific LSTM, computed over the June 2019 – June 2020 testing period. Each member was trained for ten epochs, except for five that were trained for 100 epochs. These five are marked with an additional black ring.

| Layer | Input | Activation | Output |
|---|---|---|---|
| LSTM layer 1 | (28,23) | ReLU | (28,50) |
| LSTM layer 2 | (28,50) | ReLU | (28,50) |
| LSTM layer 3 | (28,50) | tanh | (1,50) |
| dense | (1,50) | linear | (1) |

**Table 4.** Description of the LSTM architecture used in this study.

There are three stacked LSTM layers, one input and two hidden, each with fifty neurons, and a single neuron dense output layer. Following Kratzert et al. (2018) and other hydrological-LSTM papers, we use all surface or near-surface variables available in ERA5 that are potentially relevant to streamflow prediction. This gives us a total of 23 input variables, outlined in Tab. 5. For each of the ten gauges, all variables except streamflow and the historic mean (which are taken as point data at the gauge location) are averaged over the catchment (see Fig. 1) at a six-hourly frequency. We use a seven-day sequence for the input, giving the vector for each variable a length of 28 timesteps.

The LSTM was trained separately for each gauge over the training period (January 1990 to June 2018). For each gauge, the LSTM was trained 100 times using random starting weights over ten epochs each. Fig. 3 shows the training, validation, and testing of one ensemble member for the NFSW4 gauge – demonstrating that both the diurnal and seasonal cycles of this

| Surface meteorology (ERA5/IFS) | Surface hydrology (ERA5/IFS) | Subsurface hydrology (ERA5/IFS) | Other |
|---|---|---|---|
| 10-m u wind | precipitation rate | subsurface runoff | raw streamflow (GloFAS-ERA5/GloFAS) |
| 10-m v wind | total runoff | soil moisture layer 1 | bias-corrected streamflow (GloFAS-ERA5/GloFAS) |
| 2-m temperature | surface runoff | soil moisture layer 2 | mean historical streamflow for day of year (obs) |
| 2-m dewpoint | skin reservoir content | soil moisture layer 3 | |
| skin temperature | snowfall | soil moisture layer 4 | |
| | snow depth | soil temperature layer 1 | |
| | snow cover | | |
| | surface latent heat flux | | |
| | evaporation | | |

**Table 5.** List of the 23 input variables used during the training, testing, and operational use of the LSTM streamflow model.

high-volume gauge are well simulated by individual ensemble members. Five additional ensemble members were also trained over 100 epochs to determine whether extending the number of epochs provided a significant performance gain. Fig. 4 shows the performance of each ensemble member of the LSTM for each gauge, computed over the testing period (June 2019 to June 2020). We see that there is no significant advantage to extending the number of epochs beyond ten per member. The sensitivity of model skill to initial weights is quite large, in agreement with earlier studies (Kolen and Pollack, 1990; Graves et al., 2013), and could be reduced in future work by using regularisation techniques such as weight decay.

For the operational product, the mean is taken from the five best-performing ensemble members (those with the highest NSE in Fig. 4). For eight out of the ten gauges, this yields a better KGE over the testing period than using the best-performing single member, as we will see in Sec. 5.1. The LSTM is trained using data from ERA5 and GloFAS-ERA5, which we also use for the testing phase. Although this leaves the operational LSTM vulnerable to biases in the IFS, these are mitigated to an extent with the IFS being the driving model for ERA5. On the positive side, this means that evaluation of the ability of the forecasts will be conservative and thus more informative to potential users who, in not having access to decades of archived forecasts with a consistent model version, must also resort to using reanalysis training data.

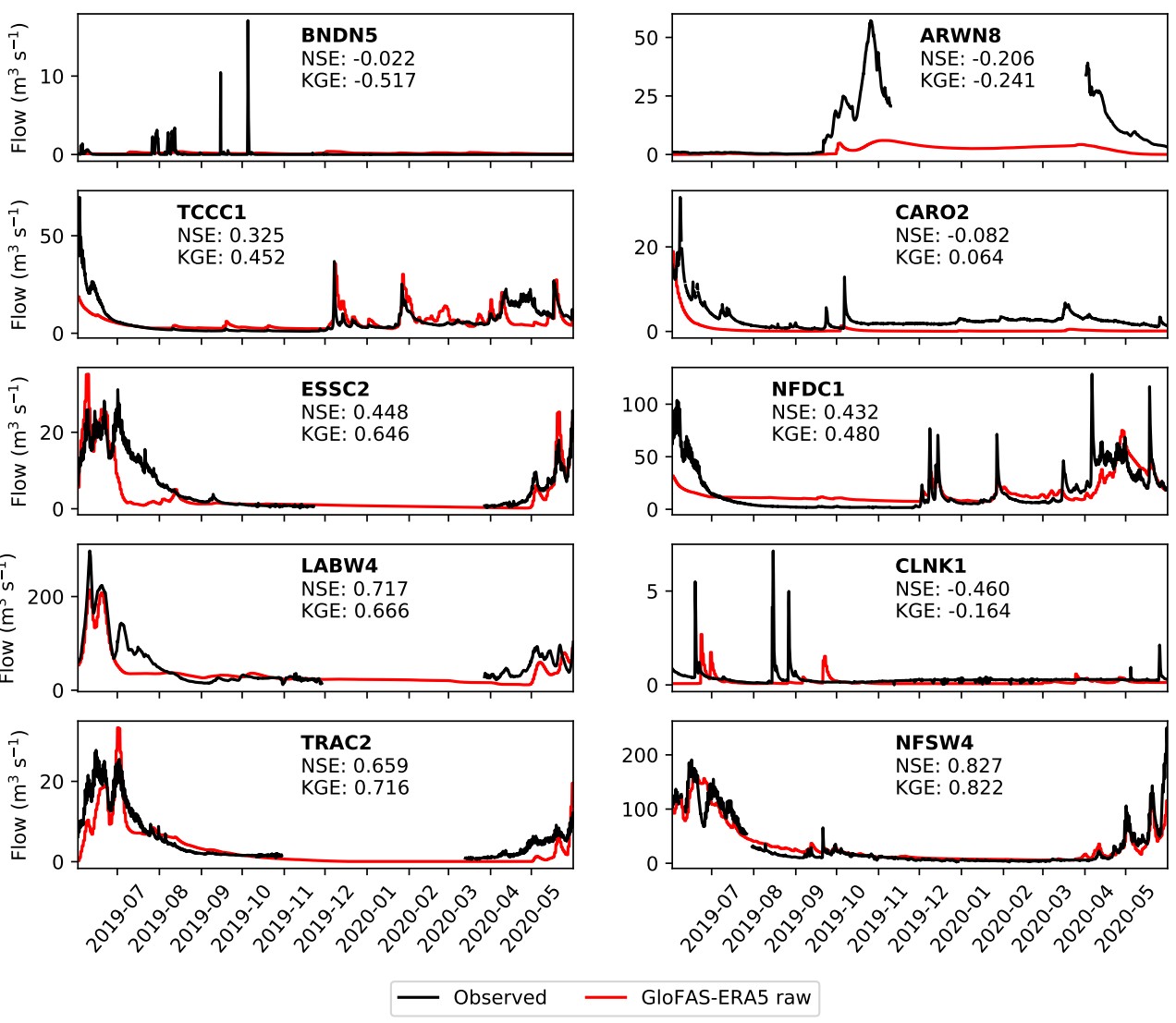

**Figure 5.** Comparison of observed streamflow [black] with raw GloFAS-ERA5 output [red] for each of the ten gauges over the testing period (June 2019 to June 2020). Gaps in the observational record over the winter are due to river freezing. Nash-Sutcliffe and Kling-Gupta efficiencies over the year-long period are given for each gauge.

## 5 Results

### 5.1 Evaluation over test period: June 2019 to June 2020

#### 5.1.1 Raw GloFAS-ERA5

Figure 5 compares raw (i.e. not bias corrected) GloFAS-ERA5 streamflow to observations at each of the ten stations over the test period (June 2019 to June 2020). Note that for the test period, all models (including in Secs. 5.1.2 and 5.1.3), are driven with ERA5 data and output a single streamflow value per gauge per timestep. The product is skilful (NSE$>0$ and KGE$> 1-\sqrt{2}$) at six of the ten stations and highly skilful (KGE$> \sqrt{2}/2$) at two. The stations where raw GloFAS-ERA5 was not skilful (BNDN5,

ARWN8, CARO2, and CLNK1) are characterised by low mean flows and highly intermittent peaks. Often, peaks appear at the wrong time – as exemplified at CLNK1 during the autumn months of 2019 – or respond too slowly or too smoothly to short precipitation stimuli.

At stations where the raw GloFAS-ERA5 product is skilful but not highly skilful (TCCC1, ESSC2, NFDC1, and LABW4) are typically marked by it capturing the annual cycle well, but generally missing some or most intraseasonal variability. This

is evident in ESSC2 and LABW4, where the summer maximum and winter minimum were well captured, but autumn and spring storms were not. Those stations where the product is highly skilful (TRAC2 and NFSW4) also capture intraseasonal variability well – see for example April and May 2020 at NFSW4, where the discharge associated with two spring storms is well simulated by the model.

#### 5.1.2 Bias-corrected GloFAS-ERA5

Bias-corrected GloFAS-ERA5 is compared with observational streamflow for each of the ten gauges over the testing period in Fig. 6. The results are a substantial improvement over the raw GloFAS-ERA5 output: NSE is improved at seven of the ten stations (except BNDN5, ARWN8, TCCC1) and KGE is also improved at seven stations (except BNDN5, TCCC1, NFDC1). Following bias-correction, GloFAS-ERA5 is now skilful at seven stations and highly skilful at four.

Failures at BNDN5 and ARWN8 are mostly due to the bias correction algorithm being unable to calibrate low and sporadic

flow; although at ARWN8, the quantile mapping brings the simulated mean and variance closer to observations as desired, resulting in an improved KGE. Although the simulated flow at CLNK1 is still not skilful, it has been improved considerably by the bias correction where the spatial optimisation routine has somewhat improved the timing of the peaks. Both CARO2 and TRAC2 also saw large increases in NSE and KGE due to improved representation of intraseasonal variability.

#### 5.1.3 LSTM

Fig. 7 shows the performance of the LSTM model, ingesting ERA5 and GloFAS-ERA5, at each gauge over the testing period. It represents a step-change in model efficiency over the bias-corrected GloFAS-ERA5 output, being skilful at nine gauges and highly skilful at six. The KGE is greater than 0.9 at three stations and the NSE is greater than 0.9 at four.

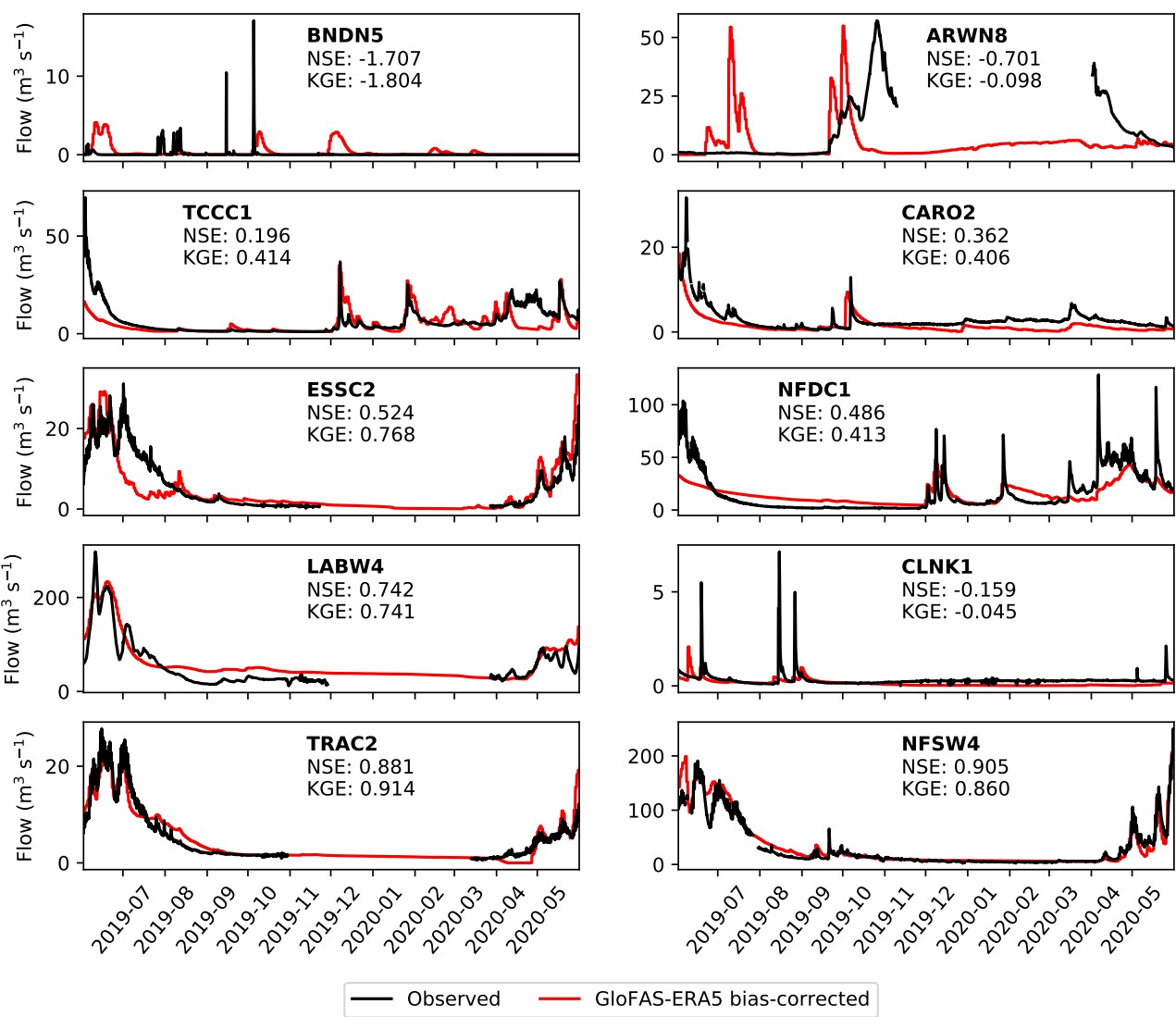

**Figure 6.** Comparison of observed streamflow [black] with bias-corrected GloFAS-ERA5 output [red] for each of the ten gauges over the testing period (June 2019 to June 2020). Gaps in the observational record over the winter are due to river freezing. Nash-Sutcliffe and Kling-Gupta efficiencies over the year-long period are given for each gauge.

Despite this, the BNDN5 and CLNK1 gauges remain relative poorly modelled (although the latter does qualify as skilful). In the observations, BNDN5 is characterised by long periods of no flow, with occasional short-lived (typically less than two days) peaks. The LSTM does manage to capture these peaks, but the timing is incorrect – usually several days late – and the magnitude is often far too small. The first of these issues, and perhaps the second, is likely due to the LSTM ingesting catchment-mean variables. The catchment basin for BNDN5 is large and arid, and therefore rain falling over it is probably not appropriately captured using this simplification. Future development of an LSTM model for streamflow at BNDN5 or similarly

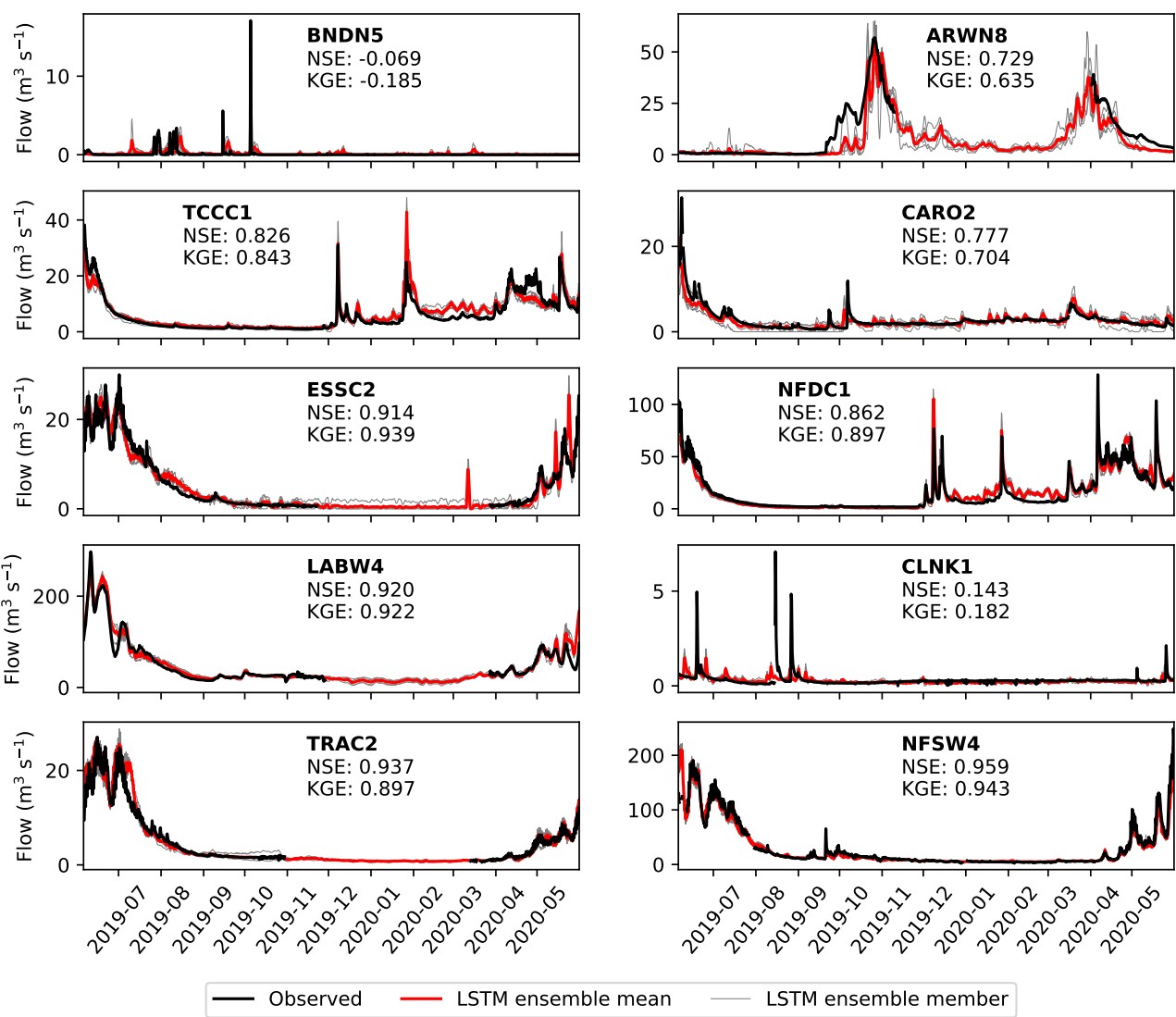

**Figure 7.** Comparison of observed streamflow [black] with the LSTM ensemble output [members:grey, mean:red] for each of the ten gauges over the testing period (June 2019 to June 2020). Gaps in the observational record over the winter are due to river freezing. Nash-Sutcliffe and Kling-Gupta efficiencies over the year-long period are given for each gauge.

large and arid basins should thus consider an expanded input that ingests variables over all (or representative) grid points in the basin. At most of the other gauges, intraseasonal variability is captured very well. This is true both for individual storms – for example high-discharge events at TCCC1 in early December 2019 and NFDC1 in late January 2020 were almost perfectly simulated – and broader flow patterns such at ESSC2 in summer 2019 and NFSW4 in spring 2020.

## 5.2 Verification over operational period: September 2020 to October 2021

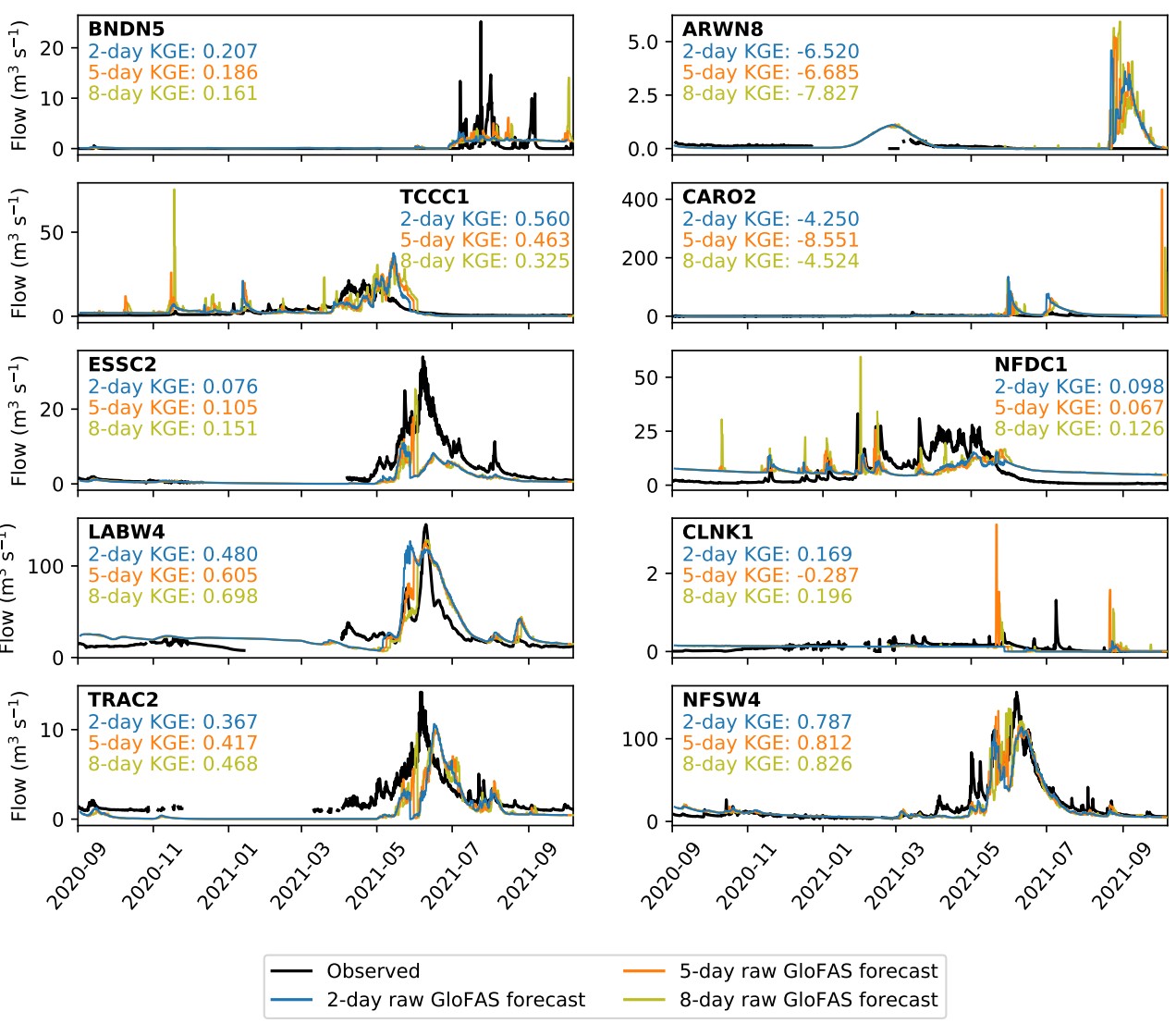

**Figure 8.** Verification of the 2-day (blue), 5-day (orange) and 8-day (yellow) raw GloFAS forecasts against observations (black) over the operational period (September 2020 to October 2021). Observations are plotted at six-hourly resolution to match the forecasts and are not plotted when the river is frozen. Kling-Gupta efficiencies for each lead time at each station are also given.

Following testing, we ran all three models (raw GloFAS, bias-corrected GloFAS, and the LSTM) operationally for thirteen
395 months between September 1 2020 and October 1 2021, producing ten-day forecasts at daily frequency. As discussed in Sec. 4, the major difference between the testing period and the operational period is the switch from ERA5 to IFS for the variables ingested by the GloFAS prediction system and by the LSTM. The effects of this switch are somewhat mitigated by using the

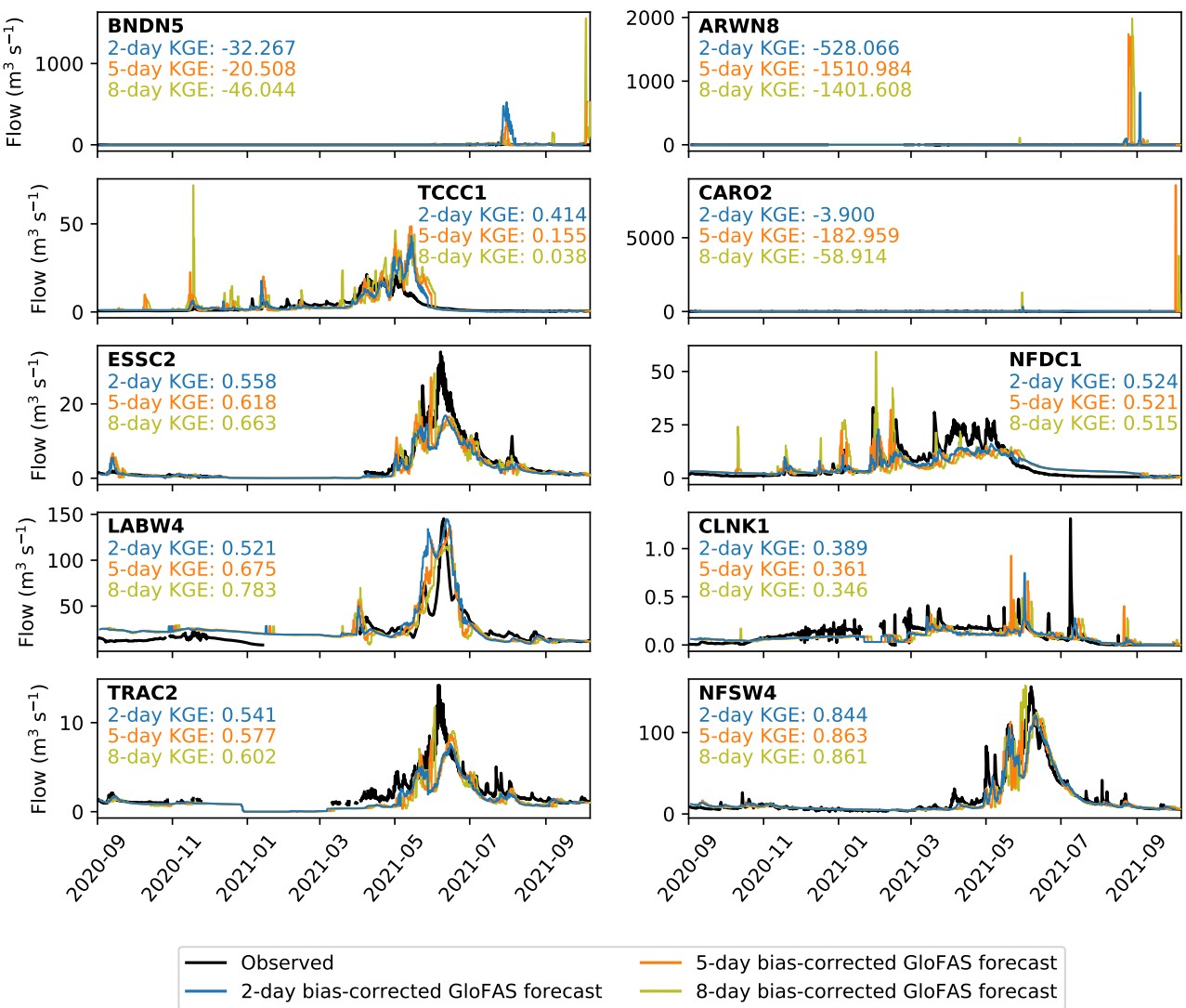

**Figure 9.** Verification of the 2-day (blue), 5-day (orange) and 8-day (yellow) bias-corrected GloFAS forecasts against observations (black) over the operational period (September 2020 to October 2021). Observations are plotted at six-hourly resolution to match the forecasts and are not plotted when the river is frozen. Kling-Gupta efficiencies for each lead time at each station are also given.

same underlying model (ERA5 uses the IFS), but there will be biases in IFS forecasts that are not present in ERA5, since the latter is nudged with simultaneous observations that are not available to the forecasts at nonzero lead times.

Forecasts at each station are evaluated at 2-, 5-, and 8-day lead times against the official observations. Results are given over the entire operational period for the raw GloFAS forecasts in Fig. 8, the bias-corrected GloFAS forecasts in Fig. 9, and for the LSTM forecasts in Fig. 10. As expected, even at short lead times, the forecasts generally perform more poorly than their reanalysis-based counterparts. There are a handful of notable exceptions to this, particularly for the raw GloFAS forecasts at

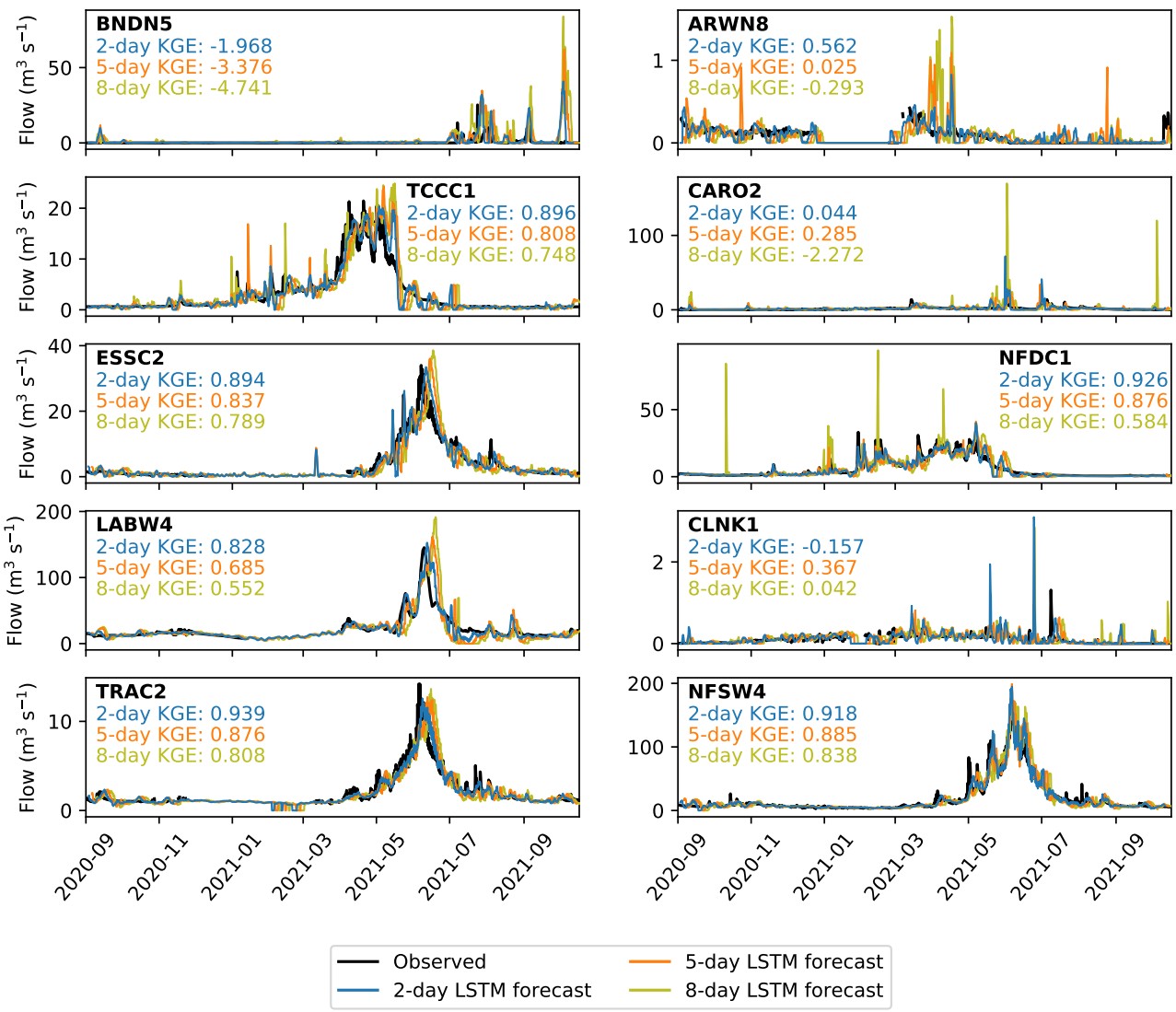

**Figure 10.** Verification of the 2-day (blue), 5-day (orange) and 8-day (yellow) LSTM forecasts against observations (black) over the operational period (September 2020 to October 2021). Observations are plotted at six-hourly resolution to match the forecasts and are not plotted when the river is frozen. Kling-Gupta efficiencies for each lead time at each station are also given.

BNDN5, TCCC1, and CLNK1. Although the GloFAS forecasts derive from a marginally more recent version of the IFS than does ERA5, the gains in this context are likely to be insignificant. At BNDN5 and CLNK1, the difference appears to be due to the hydrographs of the testing and operational periods having different characteristics. During the testing period, there was no flow at BNDN5 except for several very short (∼daily) pulses of nonzero discharge in autumn 2019; during the operational period, however, autumn 2021 was parked by a period of low but continuous flow. The lagged autocorrelation of the latter situation almost invariably makes it easier to model. At CLNK1, constant very low flow is punctuated by occasional, short-

lived peaks. There was only one significant peak in the operational period compared with four in the testing period, making it easier to model correctly. This is arguable also the case at TCCC1, where increased subseasonal variability in the operational period – notably two big storms in December 2019 and January 2021 – made it more challenging to simulate than the testing period.

Evaluating overall performance by computing the KGE of the 5-day forecasts, we find that the bias-corrected GloFAS
forecast beats the raw GloFAS forecast at six stations, the LSTM forecast beats the bias-corrected GloFAS forecast at all ten stations and the raw GloFAS forecast at nine. The raw GloFAS 5-day forecasts were skilful at seven stations and highly skilful at one of these; the bias-corrected GloFAS 5-day forecasts were also skilful at seven stations and highly skilful at one of these – though with significant improvement over the raw GloFAS forecasts at six of the seven showing skill. The LSTM 5-day forecasts were skilful at nine stations and highly skilful at five of these.

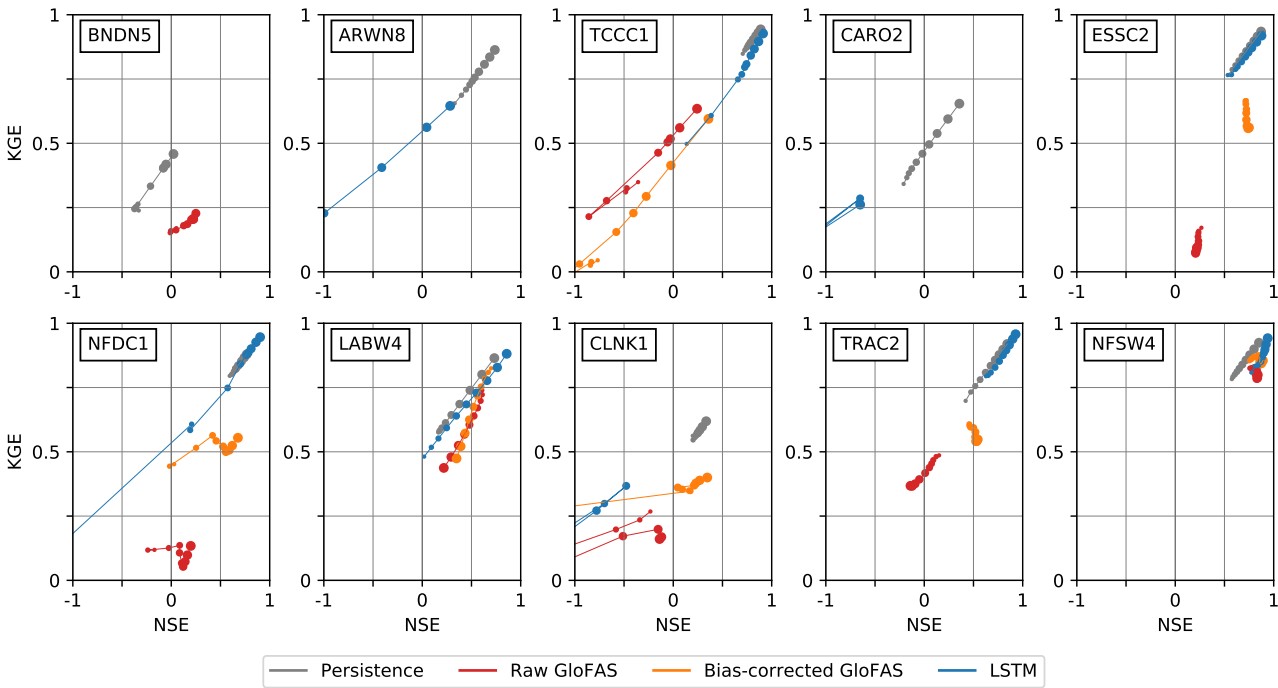

**Figure 11.** Overall performance of the three models during the operational period (September 2019 to October 2020) compared with a persistence benchmark (grey). For each model, at each station, for each lead time (from one to ten days), the Nash-Sutcliffe and Kling-Gupta efficiencies – compared to observations – are plotted. Lead times are connected sequentially by markers of diminishing size, with the ten-day lead time having the smallest marker. Values of KGE below zero or NSE below -1 are not plotted.

To complete this evaluation, we now extend our analysis to all lead times and include a simple persistence model as a benchmark (Sec. 4). For sufficiently long verification periods, the mean and variance of persistence forecasts by definition asymptotically approaches those of the observations. This has two key implications: firstly, since KGE is sensitive to the errors in forecast mean and variance, the persistence model receives a large advantage in this score compared to the other models;

secondly, following on from the first point, KGE is no longer a fair descriptor of model performance, so we must also consider
NSE.

Fig. 11 shows how the KGE and NSE vary for each model at each station as a function of lead time. Here, we are more interested in the higher values of these metrics (i.e. if a model is skilful, how skilful is it?) than the orders of magnitude of negative KGE and NSE values produced by useless forecasts, and so we truncate the plots at NSE=-1 and KGE=0. For each model at each station, scores are plotted from lead times of one to ten days inclusive, denoted by markers of diminishing
radius, and connected in order by thin lines of the same colour, creating 'phase space caterpillars'. Presenting the metrics in this way allows a quick intercomparison between models, for example we can see clearly where bias correction has worked well (ESSC2, NFDC1, TRAC2) by identifying where the bias-corrected points have moved significantly upward and to the right compared to their raw counterparts. With the exception of BNDN5, CLNK1 and CARO2, we see that the LSTM forecasts tend to have comparable KGE to the persistence forecasts, but higher NSEs. Both the persistence and LSTM forecasts tend
to outperform the raw and bias-corrected GloFAS forecasts. We also note that stations with a high lagged autocorrelation (i.e. those stations where the persistence model does well) tend also to have discharge that is well simulated by all three of the operational models. Exemplified by LABW4 and NFSW4, these are typically high-discharge sites with large catchment areas and slow response times. Indeed, those stations where the GloFAS and LSTM models struggle the most are characterised by flash responses, with their dominant mode of variability on the diurnal, rather than annual or semiannual timescale.

Next, to understand how the models perform as a function of forecast lead time, we decompose the KGE values into their three components: correlation (Fig. 12, bias ratio (Fig. 13), and variability ratio (Fig. 14). As before, we truncate the graphs to remove poor performing metric values where necessary. In terms of the correlation coefficient, the LSTM performs best at five of the ten stations (ARWN8, TCCC1, NFDC1, TRAC2, NFSW4) and most notably is the only forecast to have a positive correlation with the observation at ARWN8. This is mainly due to a fictional peak in the raw GloFAS which is exacerbated
by the bias correction, but which is filtered out by the LSTM. In addition to these five stations, the LSTM also has the highest correlation at short lead times at LABW4 but this decreases at longer lead times, dropping below the correlation coefficient of the other two forecasts after four days. The timing of the largest peak at LABW4 during the operational period is better predicted by the raw and bias-corrected GloFAS forecasts at longer lead times, whereas for the LSTM it is shorter lead times that accurately predict this peak. The raw forecast has comparable correlation coefficients to the bias-corrected and LSTM
forecasts at the three largest stations (BNDN5, CARO2, LABW4) but is notably lower for smaller catchments supporting the need for caution when using the raw GloFAS forecast for catchments below the recommended 2000 km$^2$ threshold. Since the bias-correction method is based on quantiles it can impact the correlation of the forecasts. The bias-corrected GloFAS has an improved correlation compared to the raw GloFAS forecast at seven of the ten stations (TCCC1, ESSC2, NFDC1, CLNK1, TRAC2, NFSW4, ARWN8 (not shown)).

Surprisngly, the bias ratio of the bias-corrected GloFAS forecasts is worse than than the raw GloFAS forecasts at four of the ten stations (BNDN5, ARWN8, TCCC1, CLNK1). These four stations tend to have low streamflow variability except for some short-duration large peaks. As these peaks are largely seasonal this vulnerability in the bias correction technique is likely due to the change in discharge distribution throughout the year, a problem that could be rectified by applying a season-based

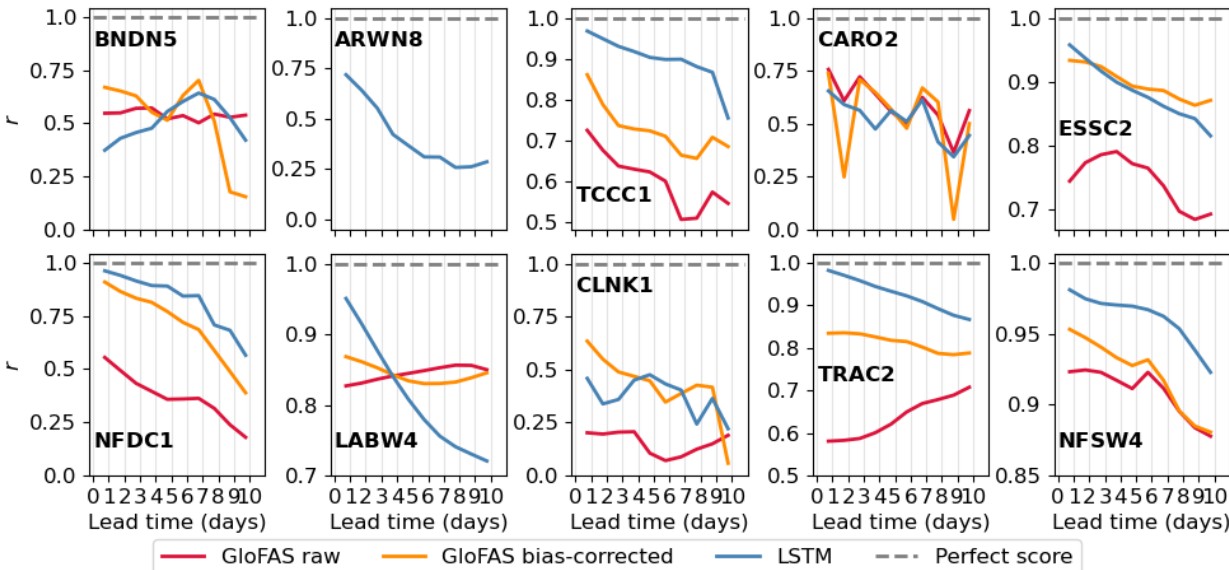

**Figure 12.** Correlation coefficient, $r$, calculated over the verification period for all three forecast models at each station for each lead-time (from 1 to 10 days). Negative correlations for ARWN8 have not been plotted.

quantile mapping. The large bias ratios at BNDN5 and ARWN8, and CLNK1 and CARO2 at longer lead-times, are often due to
unprecedentedly large values forecast by the raw GloFAS. The quantile mapping extrapolates these quantities to unphysically large values (see, e.g., 5000 m$^3$ s$^{-1}$ in September 2021 at CARO2 in Fig. 9). The raw GloFAS forecasts underpredict the streamflow at the small catchments at high elevations (ESSC2, TRAC2). The bias correction does partially correct for this bias, but the LSTM forecast still has the lowest bias at these stations. In fact, the LSTM is the least biased forecast for eight of the ten stations, the exceptions being BNDN5 and CLNK1, where the raw forecast is better. For the largest catchment, BNDN5,
the raw GloFAS forecast has a relatively small bias ratio compared to the other two forecasts mainly because it consistently predicted the zero-flow during the first half of the verification period.

The raw GloFAS forecast has the best variability ratio of all three forecasts at the two larger high-elevation catchments (LABW4, NFSW4), although at both stations the bias-corrected forecast has similar variability ratios at some lead times. However, the bias correction only improved the flow variability of raw GloFAS forecasts at half of the stations (BNDN5,
ESSC2, NFDC1, CLNK1, TRAC2). However, these stations vary significantly in catchment characteristics (catchment size, elevation, peak duration, meteorological regimes), as do the stations where the bias correction is beneficial, so there is no obviously favourable catchment characteristic. The unphysical streamflow predictions seen at BNDN5, ARWN8, CLNK1 and CARO2 also impact their variability ratio metric. The LSTM has the best variability ratio at three of the ten stations (BNDN5, ARWN8, TRAC2) and comparable values at a further four (CARO2, ESSC2, LABW4, CLNK1). All forecasts have a higher
variability ratio at longer lead-times at NFDC1 with the rate of increase similar for all three forecasts. This suggests that the

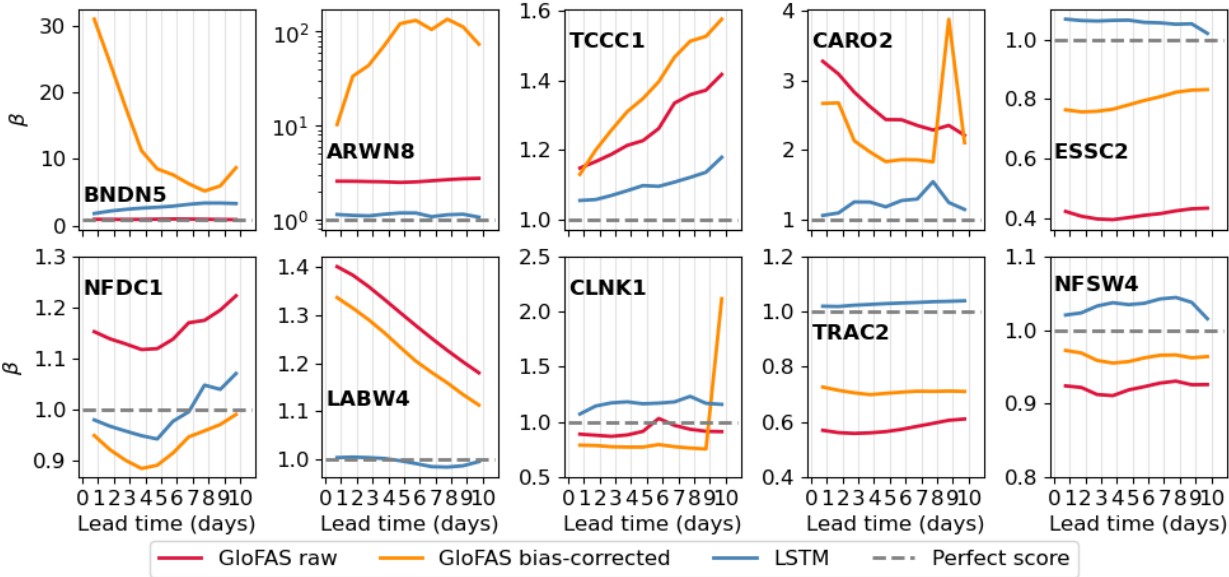

**Figure 13.** Bias ratio, $\beta$, calculated over the verification period for all three forecast models at each station for each lead-time (from 1 to 10 days). Note the logarithmic scale of the y-axis for ARWN8.

change with lead-time is due to factor impacting both such as a bias in the IFS forecast which is used to drive the LSTM and the LISFLOOD hydrological model used to create the GloFAS forecasts.

Finally, as this is a study on forecasting, we would be remiss not to analyse a single forecast. Fig. 15 shows the forecasts issued for the ten stations on May 1 2021, along with the persistence forecast, verified against observations. We see more easily here how the bias correction often – but not always – nudges the raw GloFAS forecast in the right direction, with particularly successful results at ARWN8 for this forecast. Of particular interest during this period was a pair of rain-on-snow events that impacted stations in the Rockies (NFSW4, LABW4, ESSC2, and TRAC2). These resulted in large spikes in the streamflow, visible in the observations, centred on May 3 and May 9. Neither the raw nor the bias-corrected GloFAS forecasts captured this, nor did a number of other operational forecasts (Kenneth Nowack; personal communication). While the LSTM mostly underestimated the magnitude of these events, it did predict them. This highlights the ability of LSTMs in general to learn complex non-linear relationships that may be altogether absent from physics-based models.

## 6 Discussion

### 6.1 Potential improvements

As with any newly-developed forecast models or techniques, we encountered scope for improvement during the operational phase and post-operational analysis. There are two potential improvements to the bias correction, in its current form. The first

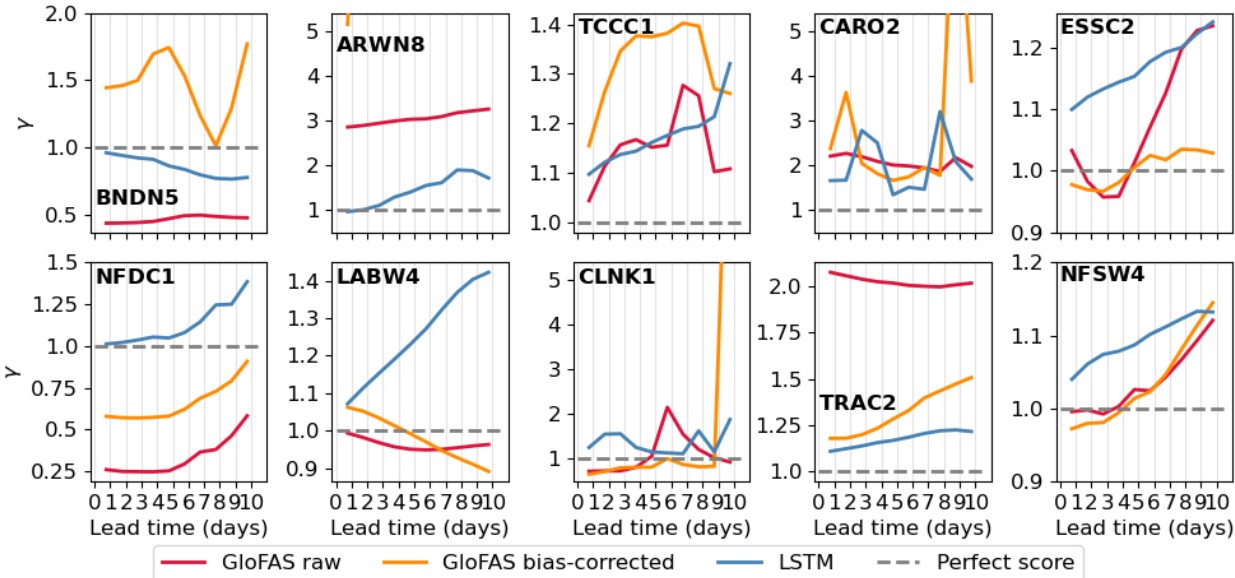

**Figure 14.** Variability ratio, $\gamma$, calculated over the verification period for all three forecast models at each station for each lead-time (from 1 to 10 days). The y-axis has been truncated to $\gamma$-values $= 5.2$ for stations ARWN8, CARO2, and CLNK1.

is to further granularise the process, computing a new quantile mapping and bias-correction matrix for (e.g.) each season. This would have the advantage of further reducing the relative bias caused by seasonally-varying environmental factors (e.g., snow on ground in winter; or intense storms in the summer saturating the soil). Such work must be careful to avoid overfitting the matrices given the increased degrees of freedom.

The bias correction would also benefit from a dependence on forecast lead time. Since biases invariably grow as a function of lead time, the correction required for a 10-day forecast is likely to be larger than the correction required for a 2-day forecast. In using GloFAS-ERA5 to compute our bias correction terms in Sec. 4.3, we effectively limited ourselves to a 0-day lead time correction. The hindcasts that would be required for this now exist from 1997 onwards, and work at ECMWF has used them for bias correction of flood threshold forecasts through CDF mapping (Zsoter et al., 2020).

Similarly, the LSTM was trained on ERA5 data, but then ingested IFS output when run operationally. Although the two products will share some biases, they will inevitably be larger in IFS (the forecast) than ERA5 (the reanalysis), resulting in errors that propagate non-linearly through the LSTM. Originally, we chose to train the model on ERA5 so that our methods were reproducible, but there is no reason other forecasters should be bound by this desire. The optimal strategy is to train the LSTM on IFS hindcasts – though this would require careful adjustment of the architecture to account for different lead times.

Similarly, such an approach must be careful of changing hindcast model versions.

     In our model, the LSTM is trained on, and subsequently ingests, catchment-mean variables (with the exception of streamflow, which is taken at the point of interest). However, as we know, rain falling far away from the station takes longer to reach it than rain falling nearby – a relationship lost when this approximation is made. By replacing the catchment-mean pre-processing step

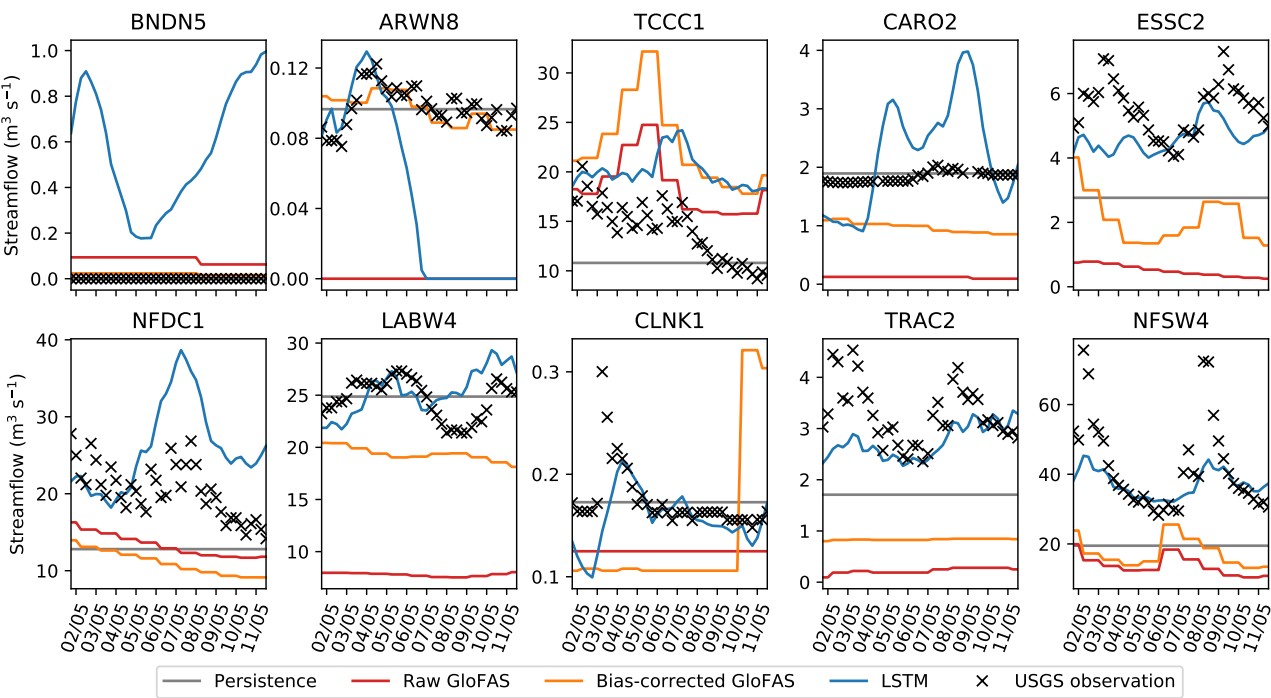

**Figure 15.** Example forecasts, issued on May 1 2021, for lead times of up to ten days. The four models (persistence: grey, raw GloFAS: red, bias-corrected GloFAS: orange, LSTM: blue) are verified against the official observations, plotted in black crosses.

with a convolutional layer in the LSTM, the LSTM would be able to learn such spatiotemporal relationships and likely produce improved forecasts as a result. As we saw in the introduction, work by Le et al. (2019) showed that even a relatively simple LSTM that ingested data from different points upstream could produce excellent results. Despite the potential advantages, there are some caveats to adding a convolutional layer. Firstly, training convolutional LSTMs is computationally very expensive. Secondly, different basins have different areas, and therefore have to ingest a different number of vectors. This either requires a new architecture for each basin (essentially unfeasible) or intelligent preprocessing, e.g., grouping the data by distance to the station. Thirdly, related to the second point, it does not allow transferability between products of different spatial resolutions without additional preprocessing. One option may be dimensionality reduction of the 2D catchment into a generalised 1D representation. Another, recently explored by Feng et al. (2021), is to use graph convolutional networks.

Could the LSTM be replaced with a simpler statistical or machine learning model which is easier to train, such as a multiple linear regression, random forest, or XGBoost? Although this may be tempting, recent research suggests that LSTMs comfortably outperform both random forest models (Adnan et al., 2021) and XGBoost (Gauch et al., 2021b) on streamflow prediction problems, particularly with increasing sample size. LSTMs are now well established as a powerful tool for modelling complex and nonlinear hydrological systems (Gauch et al., 2021a). However, their limitations – such as lack of parallelisation and long-

range dependencies – mean that they are being increasingly overlooked in sequential data problems in favour of gated recurrent units (GRUs) and attention-based models (Vaswani et al., 2017). Little research has yet been done into whether such models are more useful than LSTMs in hydrological contexts. It is likely, however, that switching to the method proposed in Kratzert et al. (2019b) – that is, using a single LSTM trained across multiple basins – would improve the forecasts. This approach would also allow the LSTM to be applied in ungauged catchments.

Generalisation when provided with out-of-distribution data and lack of interpretability are often cited as limitations of machine learning, particularly in the environmental sciences. However, as shown in Kratzert et al. (2019a), LSTMs trained across multiple basins actually perform well on previously unseen (even ungauged) basins that might typically be considered out-of-distribution. Similarly, such LSTMs can capture extreme values of streamflow driven by out-of-distribution extreme precipitation events (Frame et al., 2022). Taken together, these results suggest that such LSTMs are capable of capturing the underlying hydrological relationships that connect precipitation, runoff, and streamflow. Since climate change does not affect the underlying physical relationships, well-constructed LSTMs – and other ML models in general – should be largely immune. However, there are other types of changes, e.g. increasing urbanisation, that will affect the underlying relationships, and would then degrade the skill of the LSTM — although this would happen in any other kind of hydrological model if not appropriately updated. Although recent work has shown potential for interpretability in streamflow LSTMs (e.g., using attention theory; Li et al., 2021), this is still generally a weakness compared to physics-based models. However, in not having to rely on prescribed relationships, machine learning products can potentially learn new ones.

The LSTM as set-up in this paper still requires a hydrological model, LISFLOOD, to be calibrated in an offline procedure and then run daily in the operational period to create the forecasts. Therefore, whilst the LSTM ingests both meteorological and hydrological variables, and therefore likely represents some of the hydrological processes itself, it could be seen as a post-processing method for the GloFAS forecasts. It would be interesting to see how the difference in perspective impacts users' trust in the forecasts. Additionally, the need to run the hydrological model limits some of the benefits available via the LSTM such as computational efficiency. Future work should look at the contribution made by the LISFLOOD streamflow input data to the final LSTM forecasts and whether the impact is significant enough to warrant the additional resources. Alternatively, future research could investigate incorporating machine learning techniques directly into LISFLOOD either to speed up calculations (Höge et al., 2022; Rackauckas et al., 2020; Raissi et al., 2020) or to replicate processes that are not currently modelled in LISFLOOD within GloFAS such as the impact of reservoir management.

Finally, due to data pipeline constraints, we used only the control member of the IFS and GloFAS throughout our operational deployment. Two open research questions remain: what is the best way to combine ensemble members as inputs to an LSTM; and how can ensemble members be leveraged to provide accurate uncertainty estimates in the forecasts?

## 6.2 Potential applications

We have demonstrated, as have other authors, that LSTMs show a great deal of promise for river streamflow modelling and forecasting. Given this, there are other potential applications that are not immediately obvious from this work.

| station | BNDN5 | CLNK1 | ARWN8 | CARO2 | TCCC1 | NFDC1 | LABW4 | ESSC2 | TRAC2 | NFSW4 |
|---|---|---|---|---|---|---|---|---|---|---|
| mean discharge (m$^3$ s$^{-1}$) | 0.368 | 0.142 | 8.24 | 3.28 | 20.5 | 49.0 | 26.0 | 4.47 | 1.19 | 8.55 |
| missing obs (%) | 24 | 14 | 51 | 9.0 | 1.6 | 4.1 | 37 | 22 | 33 | 19 |
| Raw KGE (testing) | -0.517 | -0.164 | -0.241 | 0.064 | 0.452 | 0.480 | 0.666 | 0.646 | **0.716** | **0.822** |
| BC KGE (testing) | <-1 | -0.045 | -0.098 | 0.406 | 0.414 | 0.413 | **0.741** | **0.768** | **0.914** | **0.860** |
| LSTM KGE (testing) | -0.185 | 0.182 | 0.635 | 0.704 | **0.843** | **0.897** | **0.922** | **0.939** | **0.897** | **0.943** |
| Raw KGE (5-day fc) | 0.186 | -0.287 | <-1 | <-1 | 0.463 | 0.067 | 0.605 | 0.105 | 0.417 | **0.812** |
| BC KGE (5-day fc) | <-1 | 0.361 | <-1 | <-1 | 0.155 | 0.521 | 0.675 | 0.618 | 0.577 | **0.863** |
| LSTM KGE (5-day fc) | <-1 | 0.367 | 0.025 | 0.285 | **0.808** | **0.876** | 0.685 | **0.837** | **0.876** | **0.838** |

**Table 6.** Summary of results by station, showing the mean discharge and fraction of missing observations during the training period; the KGE for each of the three models over the testing period; and the KGE for 5-day forecasts from each of the three models over the operational period. Values of KGE greater than 0.707 (i.e., 'highly skilful') are marked in bold. Stations are ordered according to mean model skill during the testing period, with better performing gauges towards the right.

Perhaps the most exciting is the idea propounded by Kratzert et al. (2018), that LSTMs such as the one in this study are not black boxes, rather they are often 'grey' boxes, often containing many neurons whose output can be physically interpreted. In some cases – Kratzert et al. (2018) highlighted a neuron responsible for calculating snowmelt – representing relationships already captured by physics-based models; in others – e.g., if the rain-on-snow phenomenon described earlier is produced by a single neuron – representing relationships not necessarily captured by existing physics-based models. There is, then, the potential for new hydrological relationships to be discovered through careful investigation of a well-trained LSTM.

Another potential application is to use an LSTM model – in a basin where it has a high KGE and NSE – to infill missing data in the observational record, or to extend it further back in time where reanalysis coverage permits. Continuous long-term streamflow records are useful for both climate and hydrology research, as well as the insurance industry. Similarly, because the LSTM is extremely cheap to run once trained, it could readily be applied to climate model output (either following bias correction of that data, or by using transfer learning to adjust the internal weights of the LSTM) to produce projections of streamflow over selected basins in future climate scenarios.

## 7 Conclusions

In this study, we explored the efficacy of three models at simulating, and then forecasting up to a 10-day lead time, streamflow at ten different sites across the western United States. The forecasts were then verified against official observations and compared with a benchmark persistence model. The three models were:

1. The control member of the Global Flood Awareness System (GloFAS) ensemble, a physics-based model developed by ECMWF and the Joint Research Centre of the European Commission that provides global forecasts at a resolution of $0.1° \times 0.1°$.

2. A bias-corrected version of the raw GloFAS forecast above. The bias correction technique was newly developed for this study: firstly, each pixel is corrected using a simple quantile-mapping technique, where the mapping is computed using historical observations and the reanalysis version of GloFAS, GloFAS-ERA5. Secondly, the final streamflow is estimated using an optimised linear combination of streamflow from surrounding pixels. The matrix of coefficients for this linear combination is computed by maximising the sum of the Kling-Gupta and Nash-Sutcliffe efficiencies of the output.

3. A type of recurrent neural network, known as a long short-term memory network (LSTM), the development of which was a key focus of this study. The LSTM was trained to ingest catchment-mean meteorological and hydrological variables and output streamflow at six-hourly intervals. Trained using historical ERA5 reanalyses and observations, when run operationally, the LSTM ingested forecasts from the ECMWF Integrated Forecasting System (IFS).

Each of the three models were run for a twelve-month testing period (June 2019 to June 2020), for which they used ERA5 as input, to test how well they could simulate streamflow at the ten stations. Defining skilful as having a KGE greater than -0.414 and highly skilful as having a KGE greater than 0.707, the LSTM performed best (skilful at nine stations, highly skilful at six of these, and with a KGE exceeding 0.9 at three of those), followed by the bias-corrected GloFAS (skilful at seven, highly skilful at four), followed by the raw GloFAS (skilful at six, highly skilful at two). The bias correction improved the KGE of simulated streamflow at seven of the ten stations, implying that it is better at improving the skill of already skilful simulations than adding skill to unskilful ones.

The three models were then run operationally for a thirteen-month period (September 2020 to October 2021), using forecast output from the control member of the IFS as input. Forecast efficiencies were calculated for 2-, 5-, and 8-day lead times. Again, the LSTM performed best, with 5-day forecasts being skilful at nine stations, of which five were highly skilful; followed by the bias-corrected GloFAS, with 5-day forecasts being skilful at seven stations, of which one was highly skilful; and then raw GloFAS, which also had skilful 5-day forecasts at seven stations, one of which was highly skilful, but had lower KGE at six of the seven stations showing skill. An important caveat is that seven of the ten stations (all except BNDN5, CARO2, and LABW4) had catchment areas smaller than 2000 km$^2$, the recommended lower bound for using GloFAS forecasts. However, of these seven, the raw and bias-corrected 5-day GloFAS forecasts were skilful at six (highly skilful at one). The 5-day LSTM forecasts performed better than the raw and bias-corrected GloFAS forecasts at two of the three largest catchments not BNDN5), where the comparison is most fair. The results for all stations and models are summarised in Tab. 6. There is no significant relationship between model performance and either catchment size or altitude, despite the fact that seven gauges have catchments smaller than the recommended minimum for using GloFAS. Instead, model performance generally improves with larger mean discharge, a more pronounced seasonal cycle, and more complete historical observations.

Finally, the three models were compared at all lead times and at all stations against a benchmark persistence model. The LSTM had a higher mean NSE than the persistence model at six of the ten stations – NSE is the preferred evaluation metric

here given the dependence of KGE on errors in flow mean and variance, which are zero by definition for a long period of persistence forecasts. Overall, stations with a clearly defined annual cycle and low variance about that cycle were the easiest to predict for all models, whereas stations whose variance was dominated by single storms provided the greatest challenge. The results show a promising future for LSTMs in river streamflow forecasting.

*Code availability.* All code used to generate, train, and test the models, as well as produce the figures and analysis for this paper, are available in a dedicated GitHub repository: https://github.com/kieranmrhunt/us-streamflow/. The IFS data used to drive the forecasts are freely available from ECMWF operational archive: https://apps.ecmwf.int/archive-catalogue/?class=od.

*Author contributions.* Following the CRediT taxonomy: conceptualization – KMRH and FP; methodology – KMRH; software – KMRH and GRM; validation – KMRH and GRM; formal analysis – KMRH and GRM; writing (original draft, review, and editing) – KMRH, GRM, FP, 615 and CP; visualization – KMRH and GRM; project administration and funding acquisition – FP and CP.

*Competing interests.* The authors declare no conflict of interest.

*Acknowledgements.* KMRH would like to thank Andy Wood (UCAR) and Kenneth Nowack (Bureau of Reclamation) for constructive discussions during the early phase of this project, and Ervin Zsoter (ECMWF) for helping to debug file transfer issues. The authors would also like to thank the three reviewers (Lennart Schmidt, Frederik Kratzert, and one anonymous) for their very useful discussion comments.

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
