# Peer review of "Using a long short-term memory (LSTM) neural network to boost river streamflow forecasts over the western United States"

_Hydrology and Earth System Sciences, 2022_

## Referee Comment (RC2)

Dear authors,

I read your manuscript with great interest and think it is overall clearly structured and well written, congratulations on that. I do have a few comments, which I sincerely hope that they enhance your manuscript, none of which is anything serious. I also left a couple of minor comments below. Some of these are probably enough to be answered in the reply to my review and don't need a change in the manuscript.

The only thing that I'm not 100% sure about, and which I think should be the decision of the editor, is the following: Most of the manuscript reads to me as a technical report of your experience/participation in the streamflow forecast rodeo, which I think is a wonderful thing to publish. I'm just not sure if this manuscript should be published as a technical paper therefore, or, as is, as a research article. I do not have strong feelings about this but I thought I would mention it.

I will structure my review as follows:
- First, I try to list some general points with which I struggled during my review and which I hope can guide the authors to enhance their manuscript.
- Second, I list a couple of line-by-line comments. Some of which might need a change in the manuscript, for others it might be enough to discuss this here in the review.

Looking forward to an interesting discussion,
Frederik Kratzert

General points:
- Data and model setup: I struggled a bit to follow the exact model setup and which data, in which temporal frequency, is used in the LSTM. Some of the information is already there, but distributed across the manuscript. I think it could increase the readability to have a dedicated paragraph (e.g. in Sect. 4.3, where some of the info is already available) that simply states: "We use A, B, …, C as model input at a temporal resolution of D. The model is trained to predict E days (hours?!) of streamflow forecast, using F features during the forecast period.". Again, some of the information is already there, but distributed across different sections (e.g. Data and Model sections). I found myself multiple times searching back-and-forth in the manuscript, to try to understand exactly what you do. E.g. in line 155 you say that streamflow was available in 3-hourly resolution, L. 248f states that all (gridded) data was averaged over the catchment "at a six-hourly frequency". From looking at the table on page 11 and L 249 of the manuscript, I can see that your input data frequency was 6 hourly, but the same table also says that you have a single model output. How exactly are you generating the forecasts for all $n$ forecast timesteps?
- As one of the authors of some of the LSTM literature that you cite, I feel like I need to comment on your general training setup. It seems like your setup was mainly guided by our first LSTM paper (2018), rather than any of the more recent papers (which you cite as well). You are probably aware that many things have changed since then and that your modeling setup does not really represent the best practices when working with

LSTMs. For example, many studies (not only our papers) have shown that LSTMs really excel, when being trained on data from multiple basins at once and not on a gauge-by-gauge level as done in this manuscript. Even if the modeler is only interested in one (or a few) basin(s), it is better (with respect to the performance in the basin(s) of interest) to train on a larger set of basins. I think if you want to avoid running additional experiments, it would make sense to include at least a discussion on this topic.

- Similarly, and because you mention the computational expenses of training LSTMS, a comment on your model architecture: I know I used stacked LSTMs myself in the past (in the 2018 paper) but I was young and inexperienced. Since then, I spent many hours comparing different architectures and I never found a setup where it was worth using more than a single LSTM layer (which is what we have done since then in all of our publications). This is not critical and I don't want you to rerun experiments with different architectures, but it might be worth thinking about that when commenting on the computational expense. My experience from multiple large-sample model intercomparison studies is that the LSTM is by far the fastest model to train, compared to optimizing any hydrology model on scale. And the speed increases drastically if you use a single LSTM layer vs 3-stacked layers.

Line-by-line comments:
- L. 43 "Schmidhuber et al. 1997" missing in the Reference section and "Hochreiter and Schmidhuber 1997" is wrongly formatted in the Reference section. Usually only "Hochreiter and Schmidhuber 1997" and "Gers et al. 2000" are cited as a reference for LSTMs.
- L 48 "Chung et al. 2014" No need to cite this paper. This problem was the reason the LSTM was invented and is already discussed in "Hochreiter and Schmidhuber 1997", which would be a more appropriate reference here.
- L 57 "..even if the models were trained on multiple basins…" This is linked to my comment above. It is not "*even*" but "*because*" the models were trained on multiple basins. There are multiple studies (some of which you already cite) that show that the LSTM is able to transfer (learned) knowledge across basins. From my experience, LSTMs trained on a per-basin level are often actually not that much different from hydrology models. Another related reference that shows how model performance of LSTMs increase with the number of basins is the "The proper care and feeding of CAMELS: How limited training data affects streamflow prediction" by Gauch, Mai and Lin (2021).
- L 59 "Extending this work, Gauch et al. (2021)..." Not really sure if I understand this sentence correctly. In Gauch et al. (2021), we extended the work from the other papers to models that are able to predict streamflow (or anything actually) on any given temporal resolution (and at multiple resolutions at the same time). What do you mean with "used a reanalysis framework to demonstrate the predictive power of LSTMs in streamflow modeling"?
- L 149 "as close as observation as possible" -> "as close to observations as possible". But is this statement even true? Does GloFAS produce simulations that are "as close as

possible" to observations? You mean as close as possible "for GloFAS" (i.e. GloFAS can't produce any better simulations) or as close as possible as "any model" can get? I would question the latter.

- L 174 I shrugged when I read this passage. Personally, I really have problems to see that anything better than the long term mean is "skilful" and this is also not what Knoben et al (2019) said. Knoben et al. (2019) say that this threshold is often used to differentiate between "good" and "bad" models but they follow this with an explanation why such a global threshold is problematic in e.g. different kinds of flow regimes. Maybe you can at least extend the discussion around these thresholds a bit.
- L 204ff Great explanation!
- L 219 Eq. (7), Can you provide some explanation why you used a combination of NSE and KGE as your optimization function.
- L 226 Eq(8) Not entirely sure if I understand what I see. Is it correct that this matrix suggests that rather than including neighboring pixels (i.e. those in direct proximity to the gauge of interest) it relies on some distant pixels in all directions of the gauge? Are some of these pixels "downstream" of the gauge?
- L 235  (equation) Is this based on (personal) experience or was this specifically tested for this study? I think it could be helpful to explain this with one sentence.
- Figure 2: Great Figure and interesting results!
- L 262 "Note: snowc not available in IFS output" Is this an artifact from the manuscript preparation or otherwise can you extend what you want to say here?
- Sect 5.1.X "Evaluation of the test period": I might be missing something but can you explain what exactly is shown as a simulated hydrograph? The model is built to provide a 10-day forecast right? So for every calendar day, you would have multiple forecasted values. Which value do you pick for these plots?
- Fig 6, 7, 8 are a bit hard to read because of the small figure size and the many overlapping lines.
- L 312 I think you want to remove the "(CHECK)".
- L 420 "training them [LSTMs] is computationally very expensive". My understanding/experience from participating in various large-sample benchmarking studies is that training LSTMs is much faster than optimizing hydrology models on a similar scale. This is probably related to your specific LSTM architecture (3-stacked layer), which by itself might not be necessary.

---

## Author Comment (AC1)

General Comments

This paper is an exciting work moving machine learning models from a research lab to an operational setting. The manuscript is quite comprehensive and compares extensively to existing operational methods, highlighting advantages and challenges associated with the machine learning model (LSTM). Overall LSTM performs better than the other methods, which is expected given the nature of the machine learning model (if we have enough data).
Reply: We thank the reviewer for their positive assessment of our manuscript, and for taking the time to evaluate it critically. Our responses to their comments below are given below, point by point, in red.

Specific comments

The manuscript claims that it has been the first time that LSTM has been used in a hybrid system to create a medium-range weather forecast. In sync with the comment - RC1 Clear distinction should be made with hybrid models. This can be achieved by infographics (pictorial representation) of different models and variables used with a bounding box illustrating what is called a hybrid system and how it varies for different kinds of models. This would also help the readers to understand the entire workflow.
Reply: We could certainly add a figure describing the workflow. This will likely be combined with reviewer 2 (Frederik Kratzert)'s request to write a clear, short summary of the overall method/workflow. However, we don't think that a figure more broadly reviewing types of hybrid forecasting systems is either necessary or in scope.

Concerning line 53 "In this regard, studies fall into two categories – either seeking to create a model capable of replicating existing streamflow observations or seeking to create a model capable of forecasting streamflow at some future time. Several highly illustrative studies approach the former topic....." Though the manuscript proposes two different categories but essentially, from a machine learning perspective, they might not be very different. Replication is also a form of prediction for a machine learning model. Distinction based on this might not be appropriate here. A rather significant difference is the use of streamflow at previous timesteps. Or in general, the first approach could be using only the drivers (precipitation, temperature radiation, wind) where we have almost no explicit information about the inherent state of the catchment (things like how moist is the soil, how much snow we have in the catchment, which might melt) to make predictions of streamflow (series of papers published by researchers at Johannes Kepler University Linz is in this direction). While the second category could be where we explicitly include the inherent state information (flow at previous time steps) about the catchment, which will have more potential for better predictions.
Reply: We agree with the reviewer that there is very little, if any, distinction between

replication and prediction (i.e. forecasting) in the research context of machine learning models. However, these are clearly different tasks when it comes to application because they solve different problems and (as the reviewer highlights) ingest different data. Therefore, we believe the distinction is very much appropriate to make here, and, given the subject of the paper is the operational application of an LSTM, it makes sense to include a discussion of previous work that delineates between earlier "theoretical" (i.e. replication) and "applied" (i.e. forecasting/prediction) experiments.

That said, the distinction between modelling basins that have, as the reviewer says, "almost no explicit information" and basins that have "inherent information" – i.e. essentially ungauged and gauged basins respectively – is interesting from a research point of view and we are happy to include a sentence or two discussing this in the revised introduction.

For the LSTM model, 23 variables have been chosen for predictions; while not doing any extensive hyperparameter search (optimum number of variables), some rationale needs to be provided on why those variables were chosen (if any kind of qualitative selection was made).
Reply: Following Kratzert et al (2018) section 2.4, we included all surface or near-surface variables available in ERA5 that were potentially relevant to streamflow prediction. We will clarify this in the revised manuscript.

Figure 2 of the paper is really interesting, and we can see that for some catchments, through different epochs NSE (and RMSE) changes a lot for models initialised with different weights. Is it normal for all machine learning models to vary a lot after hyperparameter tuning?
Reply: Thank you. Yes – this has been an active field of study for some time (e.g. Kolen and Pollack 1990). There is also evidence that this is the case for RNNs (e.g. Graves et al., 2013).

Kolen, J., & Pollack, J. (1990). Back propagation is sensitive to initial conditions. Advances in neural information processing systems, 3.

Graves, A., Mohamed, A. R., & Hinton, G. (2013, May). Speech recognition with deep recurrent neural networks. In 2013 IEEE international conference on acoustics, speech and signal processing (pp. 6645-6649). IEEE.

Secondly, we also see that the variability in NSE (and RMSE) is also high when models are trained for 100 epochs, and they perform worse than the best models trained with 10 epochs. This could also result from overfitting. While the figure provides a nice way to represent the uncertainties associated with the model, it might also make them look more uncertain than they actually are. As there are methods in machine learning to decrease this variability, it would be interesting to

see how models perform if trained for 100 epochs with early stopping criteria.
Reply: The reviewer is correct. Following this comment and several on similar lines from reviewer 1 (Lennart Schmidt), we will improve the training discussion in the revised manuscript, including a new figure on loss by epoch.

The discussion section is focused on the use of the convolution layer, a big challenge would be handling the different sizes of catchments. Generalised 1D representation of 2-D values might be a research direction. The other could be the use of graph neural network. A good example could be in the direction of the paper "Spatial and Temporal Aware Graph Convolutional Network for Flood Forecasting"
Reply: Thanks for the suggestion and reference – happy to add these to the revised discussion.

Feng, Z. Wang, Y. Wu and Y. Xi, "Spatial and Temporal Aware Graph Convolutional Network for Flood Forecasting," 2021 International Joint Conference on Neural Networks (IJCNN), 2021, pp. 1-8, doi: 10.1109/IJCNN52387.2021.9533694.

Minor technical comment :

Figure 2 A suggestion would be to either create a plot with a colour gradient for the density of points, else making the marker size smaller can help in illustrating where most of the points are (reducing the overlap).
Reply: OK – we're happy to play around with point size here to try and improve the figure's clarity.

---

## Author Comment (AC2)

Dear authors,

I read your manuscript with great interest and think it is overall clearly structured and well written, congratulations on that. I do have a few comments, which I sincerely hope that they enhance your manuscript, none of which is anything serious. I also left a couple of minor comments below. Some of these are probably enough to be answered in the reply to my review and don't need a change in the manuscript.

The only thing that I'm not 100% sure about, and which I think should be the decision of the editor, is the following: Most of the manuscript reads to me as a technical report of your experience/participation in the streamflow forecast rodeo, which I think is a wonderful thing to publish. I'm just not sure if this manuscript should be published as a technical paper therefore, or, as is, as a research article. I do not have strong feelings about this but I thought I would mention it.

I will structure my review as follows:
- First, I try to list some general points with which I struggled during my review and which I hope can guide the authors to enhance their manuscript.
- Second, I list a couple of line-by-line comments. Some of which might need a change in the manuscript, for others it might be enough to discuss this here in the review.
Looking forward to an interesting discussion,
Frederik Kratzert

Thank you for taking the time to review our manuscript and for providing us with some very helpful suggestions, from which we believe the manuscript will benefit greatly. We have responded to these points individually below, in red.

General points:
- Data and model setup: I struggled a bit to follow the exact model setup and which data, in which temporal frequency, is used in the LSTM. Some of the information is already there, but distributed across the manuscript. I think it could increase the readability to have a dedicated paragraph (e.g. in Sect. 4.3, where some of the info is already available) that simply states: "We use A, B, …, C as model input at a temporal resolution of D. The model is trained to predict E days (hours?!) of streamflow forecast, using F features during the forecast period.". Again, some of the information is already there, but distributed across different sections (e.g. Data and Model sections). I found myself multiple times searching back-and-forth in the manuscript, to try to understand exactly what you do. E.g. in line 155 you say that streamflow was available in 3-hourly resolution, L. 248f states that all (gridded) data was averaged over the catchment "at a six-hourly frequency". From looking at the table on page 11 and L 249 of the manuscript, I can see that your input data frequency was 6 hourly, but the same table also says that you have a single model output. How exactly are you generating the forecasts for all $n$ forecast timesteps?

We agree that the manuscript could really benefit from a summary of the overall setup and operational deployment, which we will include, as suggested, in the introduction. We will also act on similar comments by the other two reviewers to clarify our methods throughout, for example including a table of input hydrometeorological variables.

- As one of the authors of some of the LSTM literature that you cite, I feel like I need to comment on your general training setup. It seems like your setup was mainly guided by our first LSTM paper (2018), rather than any of the more recent papers (which you cite as well). You are probably aware that many things have changed since then and that your modeling setup does not really represent the best practices when working with LSTMs. For example, many studies (not only our papers) have shown that LSTMs really excel, when being trained on data from multiple basins at once and not on a gauge-by-gauge level as done in this manuscript. Even if the modeler is only interested in one (or a few) basin(s), it is better (with respect to the performance in the basin(s) of interest) to train on a larger set of basins. I think if you want to avoid running additional experiments, it would make sense to include at least a discussion on this topic.
We agree with this comment. When I set up the LSTM originally a few years ago, the methodology followed your 2018 paper (which was then more-or-less the only practice, let alone the best practice). Of course, we do appreciate that accepted best practice has changed in the meantime and will gladly discuss this in more detail in the revised introduction. Given that the per-basin LSTM results are still reasonably good, we hope that the reviewer won't insist on a complete multi-basin rerun of our entire project!

- Similarly, and because you mention the computational expenses of training LSTMS, a comment on your model architecture: I know I used stacked LSTMs myself in the past (in the 2018 paper) but I was young and inexperienced. Since then, I spent many hours comparing different architectures and I never found a setup where it was worth using more than a single LSTM layer (which is what we have done since then in all of our publications). This is not critical and I don't want you to rerun experiments with different architectures, but it might be worth thinking about that when commenting on the computational expense. My experience from multiple large-sample model intercomparison studies is that the LSTM is by far the fastest model to train, compared to optimizing any hydrology model on scale. And the speed increases drastically if you use a single LSTM layer vs 3-stacked layers.
This is a very good point; one which we will gladly discuss in the revised manuscript. Regarding the benchmarking of (a) stacked vs single-layer LSTMs and (b) single-layer LSTMs vs hydrological models, are there particular references you would recommend? I was hoping to see some relevant data in your 2019 paper (https://hess.copernicus.org/preprints/hess-2019-368/hess-2019-368.pdf) but it didn't seem to touch on the issue of training speed.

Line-by-line comments:
- L. 43 "Schmidhuber et al. 1997" missing in the Reference section and "Hochreiter and Schmidhuber 1997" is wrongly formatted in the Reference section. Usually only "Hochreiter and Schmidhuber 1997" and "Gers et al. 2000" are cited as a reference for LSTMs.
Thanks – we will make these changes.

- L 48 "Chung et al. 2014" No need to cite this paper. This problem was the reason the

LSTM was invented and is already discussed in "Hochreiter and Schmidhuber 1997", which would be a more appropriate reference here.
OK – this is a reasonable substitution that we're happy to make.

- L 57 "..even if the models were trained on multiple basins…" This is linked to my comment above. It is not "*even*" but "*because*" the models were trained on multiple basins. There are multiple studies (some of which you already cite) that show that the LSTM is able to transfer (learned) knowledge across basins. From my experience, LSTMs trained on a per-basin level are often actually not that much different from hydrology models. Another related reference that shows how model performance of LSTMs increase with the number of basins is the "The proper care and feeding of CAMELS: How limited training data affects streamflow prediction" by Gauch, Mai and Lin (2021).
We will include these changes in our expanded discussion of the literature (see also response to major point 3). Thank you for the reference.

- L 59 "Extending this work, Gauch et al. (2021)…" Not really sure if I understand this sentence correctly. In Gauch et al. (2021), we extended the work from the other papers to models that are able to predict streamflow (or anything actually) on any given temporal resolution (and at multiple resolutions at the same time). What do you mean with "used a reanalysis framework to demonstrate the predictive power of LSTMs in streamflow modeling"?
We added this to demonstrate that work had been done on predicting streamflow using reanalyses, and not just observations (i.e. work deriving from the CAMELS datasets). The summary suggested in this comment is helpful though, so we will include it in our revision.

- L 149 "as close as observation as possible" -> "as close to observations as possible". But is this statement even true? Does GloFAS produce simulations that are "as close as possible" to observations? You mean as close as possible "for GloFAS" (i.e. GloFAS can't produce any better simulations) or as close as possible as "any model" can get? I would question the latter.
We definitely mean "for GloFAS" here, and will rephrase this in the revised manuscript.

- L 174 I shrugged when I read this passage. Personally, I really have problems to see that anything better than the long term mean is "skilful" and this is also not what Knoben et al (2019) said. Knoben et al. (2019) say that this threshold is often used to differentiate between "good" and "bad" models but they follow this with an explanation why such a global threshold is problematic in e.g. different kinds of flow regimes. Maybe you can at least extend the discussion around these thresholds a bit.
Yes, of course any definition of "skilful" is typically quite arbitrary. I certainly agree that just doing better than the climatology/mean doesn't really indicate meaningful skill, a definition that has irritated me in seasonal model evaluations for years. So perhaps this usage is a case of Stockholm Syndrome, but I'm happy to discuss it a bit more – including correcting the statement attributed to Knoben et al (2019) – in the revised manuscript.

- L 204ff Great explanation!

Thank you!

- L 219 Eq. (7), Can you provide some explanation why you used a combination of NSE and KGE as your optimization function.

This is a good question, for which we will include an answer in the revised manuscript. Essentially, using NSE alone leaves the optimization procedure vulnerable to incorrect local minima (e.g. incorrect mean). However, we also found that using KGE alone tended to result in a bias correction that weighted correct mean and variance too highly compared to correlation (not useful for improving forecasts). We found that combining the two improved the weighting more in favour of correlation, while avoiding spurious local minima.

- L 226 Eq(8) Not entirely sure if I understand what I see. Is it correct that this matrix suggests that rather than including neighboring pixels (i.e. those in direct proximity to the gauge of interest) it relies on some distant pixels in all directions of the gauge? Are some of these pixels "downstream" of the gauge?

Yes – this interpretation is correct. All pixels are in the close neighbourhood of the gauge, but some are indeed downstream. This might seem a bit of a strange choice, but there are two advantages: (1) it is quick to set up, no additional topology data or preprocessing is required; and (2) we think (although we have not tested it) that it could help reduce temporal biases in some cases, for example, where the model puts the flow "too early", downstream information can actually be useful in correcting the bias. Where downstream information is not useful – as in the example given for NFSW4 in the paper – the weights collapse to zero (or very close) anyway.

- L 235 (equation) Is this based on (personal) experience or was this specifically tested for this study? I think it could be helpful to explain this with one sentence.

This was specifically tested at the beginning of the operational phase when we realised that the differences between GloFAS and GloFAS-ERA5 were leading to an overenthusiastic bias correction. We will mention this in the revised methods section.

- Figure 2: Great Figure and interesting results!

Thank you!

- L 262 "Note: snowc not available in IFS output" Is this an artifact from the manuscript preparation or otherwise can you extend what you want to say here?

This is left over from the original manuscript preparation and will be removed in the revision.

- Sect 5.1.X "Evaluation of the test period": I might be missing something but can you explain what exactly is shown as a simulated hydrograph? The model is built to provide a 10-day forecast right? So for every calendar day, you would have multiple forecasted values. Which value do you pick for these plots?

All models are driven with ERA5 during the testing period, since they are not being run

in forecast mode and we simply want to test their respective abilities to replicate observed streamflow. This contrasts with Figs 6/7/8, where the models *are* run in forecast mode (driven by IFS output), and for which we plot hydrographs at three different lead times.

- Fig 6, 7, 8 are a bit hard to read because of the small figure size and the many overlapping lines.
We appreciate that these figures are quite compact – but they are vector images so readers can get finer detail by zooming in. If the reviewer thinks that removing a timeseries from each (e.g. 2-day lead time forecasts) would be helpful, we could do that.

- L 312 I think you want to remove the "(CHECK)".
Thank you for spotting this – will remove.

- L 420 "training them [LSTMs] is computationally very expensive". My understanding/experience from participating in various large-sample benchmarking studies is that training LSTMs is much faster than optimizing hydrology models on a similar scale. This is probably related to your specific LSTM architecture (3-stacked layer), which by itself might not be necessary.
This comment is very similar to the last of your general comments. Again, we will happily include this caveat in our revised discussion.

---

## Author Comment (AC3)

**General comments**

The manuscript delivers an interesting addition to the current surge of machine-learning in hydrological modelling by extending the application of LSTMs from pure streamflow modeling to actual forecasting. To do so, they ingest the output of physical forecasting systems as input to an LSTM. The main result, that LSTM outperforms the other approaches, is not surprising as the LSTM merely acts as a bias correction algorithm with many more degrees of freedom. Nevertheless, this is still a relevant finding that should be disseminated throughout the hydrological community. The manuscript is well-structured and comprehensible, particularly the introduction and methods parts are very comprehensive yet concise. Intuitive measures of model performance, a solid discussion of the error metrics as used in the study, a comprehensible discussion on the nature and choice of datasets as well as informative plots act as a solid foundation for the reader to follow along w.r.t to the methodological execution and its results. However, towards the end, the discussion and conclusion do miss out to put the work and the results into a broader perspective e.g. by contrasting it against ongoing machine-learning research, its limitations or future directions for hybrid modeling (see detailed comments below).

We thank the reviewer for giving his time to provide a detailed review of our manuscript. We respond to his points individually below, in red.

**Scientific/Specific comments**

1. Please elaborate on the different types of "Hybrid" models/forecasts that are possible. In ML literature, there are current advances, termed "hybrid models", of including and solving differential equations inside the NN, promising the best of both worlds (high accuracy while keeping interpretability/robustness to out-of-distribution cases). These developments should be listed as future directions of research and the approach of this manuscript should be contrasted against these new development in the introduction.

Rackauckas, Christopher, et al. "Universal differential equations for scientific machine learning." arXiv preprint arXiv:2001.04385 (2020).

Raissi, Maziar, Alireza Yazdani, and George Em Karniadakis. "Hidden fluid mechanics: Learning velocity and pressure fields from flow visualizations." Science 367.6481 (2020): 1026-1030. APA

We agree with this comment and will add these discussion points and references to the revised introduction so that it also covers the limitations and future directions of hybrid modelling in hydrology. Thank you for the suggested references.

2. As the title suggests, the LSTM-approach is set up to "boost" the forecasts of IFS. As the LSTM receives Era5/IFS streamflow estimates, I personally would rather view it as a more sophisticated bias correction (more parameters, less constraints) than a

separate modeling approach. I believe that the manuscript would benefit if this was put into perspective in the discussion part. Also, the possibility of using a simple statistical/ML model with less training effort, e.g. a simple linear model or RandomForest should at least be mentioned as an alternative.

We disagree with this. Although the boundary is vague and ML statistical correction, hybrid modelling, and pure ML models exist on a spectrum, the method implemented here would still work even if GloFAS were removed from the input (though perhaps with reduced skill). NWP variables are used as input, and so the LSTM is in some way replicating the hydrological processes, as in the Kratzert papers cited throughout.

3. Especially when taking into account that 7/10 catchments are known to be too small for the raw GloFAS to perform well, it is obvious that a statistical bias correction outperforms the raw forecast more the more degrees of freedom it is given. The "unfairness" of the comparison of raw GLOFAS vs. LSTM (world-wide simulation at 0.1° resolution vs. local model) should be highlighted in the introduction and discussion sections.

Yes – this is of course part of the motivation for trialling an LSTM in the first place. We will add this caveat to the revised introduction and discussions.

4. It would be beneficial to include and elaborate a bit further the motivation behind the two bias correction algorithms in the introduction to 4.2., i.e. quantile (remap values to reduce systematic bias) and spatial (inherent spatial bias in GLOFIS-ERA5(?)).

Quantile mapping is a fairly standard practice in hydrological bias correction (e.g. Thrasher *et al*., 2012). The spatial mapping provides an additional layer of bias correction, accounting for the fact that consistent spatiotemporal biases in hydrometeorological fields such as precipitation (should they exist), will result in consistent upstream/downstream biases in streamflow – information that can be used to improve forecasts. We will expand the revised methods section to include this motivation.

Thrasher, B., Maurer, E. P., McKellar, C., & Duffy, P. B. (2012). Bias correcting climate model simulated daily temperature extremes with quantile mapping. Hydrology and Earth System Sciences, 16(9), 3309-3314.

Also, the motivation for the two final steps should be explained and justified in greater detail. What do you mean by different climatologies, why split 3/4 vs. 1/4? Why do you shift the forecasts that have ben quantile mapped-once more?

This is a fair question and has also been raised by reviewer 2. GloFAS-ERA5 and GloFAS forecasts have different climatologies because they take meteorological input from different sources (ERA5 and IFS forecasts respectively). ERA5 and IFS themselves have different climatologies because, although they share the same driving model, ERA5 is a reanalysis and is nudged towards observations, whereas

IFS forecasts aren't. We realised at the beginning of the operational phase that the differences between GloFAS and GloFAS-ERA5 were leading to an overenthusiastic bias correction, and so damped it using the weights given. This point was also raised by reviewer 2, and we will expand the revised methods section to include this explanation.

Also the fact that the bias correction has been newly developed (mentioned in conclusions) should be placed in the respective chapter in the methods part.
Yes – we will mention this in the revised methods section.

5. You argue that you used reanalysis data during train+test to make the results reproducible for potential users. But are the operational forecasts using IFS still reproducible? If not, it would be beneficial to provide the respective data on zenodo or a similar platform.
The IFS data used for these forecasts is freely available from the MARS web service hosted by ECMWF. The full streamflow forecasts themselves are available on the project GitHub page.

6. Generally, the manuscript misses to give detailed information on the training process. I would advise to include loss curve(s) (loss vs. epoch) of test and train. This is the common way to present information to see whether training was successful. Also, train and test error metrics should be provided to give an intuition whether under- or overfitting might have happened. The same applies to the bias correction, here the reader is not provided with any information on the optimization procedure or performance, even though there is a risk of overfitting. Similarly, information on the loss function, training hyperparameters (dropout, decay, learning rate, recurrent activation etc.) should be listed in appendix or refer to repository. To this end, the training process could have been performed more thoroughly: Hyperparameter-Tuning usually involves searching over model/training parameters as well as model configurations (n hidden layers, nodes etc.), not only epochs. The latter is usually less relevant. Ideally, hyperparameter tuning would be executed in a cross-validation set-up. To that respect, please elaborate what "tuned using sensitivity tests" refers to (l.240).
We agree that we have provided too little detail on the LSTM training process – although we would like to note that as the code is open access, a sufficiently interested reader could find the finer details there. However, in response to this suggestion and in keeping with ML literature conventions, we are happy to expand the methods section to include more information on (a) the final choice of hyperparameters, (b) how we arrived at these choices, and (c) how we avoided overfitting.
With the bias correction, which is ultimately a linear – if somewhat multivariate – model, we can be fairly sure that overfitting has not occurred for two reasons. Firstly, the number of coefficients sought is several orders of magnitude less than the number of training data points. Secondly, the validation – for which we used

data set aside from the training/fitting process – suggested a very good fit. We will add these points to the revised methodology.

7. The conclusion is (too) detailed on the technical side (n of skilful vs. non-skilful results, KGE vs. NSE) but misses to provide the most relevant point in 2-3 comprehensive sentences: Where exactly does LSTM perform best/worst (seasonal, diurnal, altitude etc.), Where are its limitations? Both the discussion and conclusion miss to put the results into a broader perspective: How could the LSTM be improved other than switching to Convolutional LSTM-layers? What are future research directions? Maybe some points to consider here:
We agree that some more detailed discussion on limitations and future research areas for ML in hydrology would be useful, as well as a synthesis on overall performance. We respond to the suggestions individually below.

- Limitations of LSTMs: Some limitations are fairly well-acknowledged by now (parallelization, long-range dependencies) so that LSTMs are not the go-to model for sequential data anymore. Thus, outlook on new developments like GRUs and, most importantly attention-based models (e.g. transformers) should be included
  Although LSTMs do have some limitations, they have been shown to be an incredibly powerful tool when it comes to modelling hydrological systems (e.g. Gauch *et al*, 2021). Of course, as ML models become increasingly sophisticated, there is little doubt that novel architectures/technique will perform such tasks much better, but there is very little (if any) published research on that yet. We will add a short discussion of these new developments in general sequential modelling to the revised discussion.

  Gauch, M., Kratzert, F., Klotz, D., Nearing, G., Lin, J., & Hochreiter, S. (2021). Rainfall–runoff prediction at multiple timescales with a single Long Short-Term Memory network. Hydrology and Earth System Sciences, 25(4), 2045-2062.

- Limitations of ML generally: Generalisation when provided with out-of-distribution data, e.g. due to system changes (climate change), lack of interpretability
  These are indeed often cited as shortcomings of ML. However, as shown in Kratzert *et al*. (2019), LSTMs trained across multiple basins actually perform well on previously unseen (even ungauged) basins that might typically be considered out-of-distribution. Similarly, such LSTMs can capture extreme values of streamflow driven by out-of-distribution extreme precipitation events (Frame *et al*., 2021). Taken together, these results suggest that such LSTMs are capable of capturing the underlying hydrological relationships that connect precipitation, runoff, and streamflow. Since climate change doesn't affect these physical relationships (only the magnitudes of the inputs), sufficiently advanced LSTMs should be largely immune. That said, other types

of changes, e.g. increasing urbanisation, can affect the underlying relationships, and would degrade the skill of the LSTM – though this would of course happen in any other type of hydrological model if not updated. Although recent work has shown potential for interpretability in streamflow LSTMs (e.g. using attention theory; Li *et al.*, 2021), we appreciate this is still generally a weakness compared to physics-based models – although in not having to rely on prescribed relationships, ML products can potentially learn new ones.

Kratzert, F., Klotz, D., Herrnegger, M., Sampson, A. K., Hochreiter, S., & Nearing, G. S. (2019). Toward improved predictions in ungauged basins: Exploiting the power of machine learning. Water Resources Research, 55(12), 11344-11354.

Frame, J., Kratzert, F., Klotz, D., Gauch, M., Shelev, G., Gilon, O., ... & Nearing, G. S. (2021). Deep learning rainfall-runoff predictions of extreme events. Hydrology and Earth System Sciences Discussions, 1-20.

Li, W., Kiaghadi, A., & Dawson, C. (2021). High temporal resolution rainfall–runoff modeling using long-short-term-memory (LSTM) networks. Neural Computing and Applications, 33(4), 1261-1278.

- Greater picture:
    - Comparison to other hybrid modelling approaches in ML (see above)
      Yes – as with your first major comment, we are happy to expand the discussion to include greater coverage of hybrid modelling approaches.
    - How far away are we from entirely ML-based forecasts, given considerable advances in ML-based climatological forecasts? Should science still focus on improving relatively coarse physical model like ERA5 or rather explore ML-based bias correction at a large spatial scale?
      It is natural to speculate on this, but one may as well ask "why should we continue improving cars when we have planes?" Ultimately, physics-based and ML-based hydrological models should both continue to be improved since, although they can be used for the same purpose (e.g. forecasting, as in this paper), they have different strengths and weaknesses – as discussed above –and will thus continue to be suitable for different applications.

- Reflect on the risk of overfitting in bias correction + lstms, including the fact that, ideally, one would have to estimate the bias of IFS w.r.t to ERA5.

Yes – this was one of our biggest concerns, as we stated in the original discussion: "Similarly, the LSTM was trained on ERA5 data, but then ingested IFS output when run operationally. Although the two products will share some biases, they will inevitably be larger in IFS (the forecast) than ERA5 (the reanalysis), resulting in errors that propagate non-linearly through the LSTM. Originally, we chose to train the model on ERA5 so that our methods were reproducible, but there is no reason other forecasters should be bound by this desire. The optimal strategy is to train the LSTM on IFS hindcasts – though this would require careful adjustment of the architecture to account for different lead times. Similarly, such an approach must be careful of changing hindcast model versions."

Vaswani, Ashish, et al. "Attention is all you need." Advances in neural information processing systems 30 (2017).

**Technical Corrections**

- The abstract could be shortened
  We appreciate that at ~400 words, the abstract is a little on the longer side. However, the paper covers quite a wide variety of work and new methodology, so it isn't necessarily obvious how it could be shortened without loss of impact. We are happy to take guidance on this.

- l. 85, add bracket
  Thanks – we have fixed this.

- l. 110: "interesting challenge" is a subjective statement, rephrasing is advised
  We will remove "interesting" in the revised manuscript.

- l. 252 wether A extending
  Sorry – not really sure what the suggestion is here.

- l.262 remove note
  We have removed this.

- l. 281: Is it 6 or 7 catchments that are skilful? 5.1.1. lists 6/10 that are skilful, here it is "still" 7/10?
  Good spot – we have edited this in the revised manuscript so that it now says: "Following bias-correction, GloFAS-ERA5 is now skilful at seven stations and highly skilful at four."

- l. 312 remove (CHECK)
  Thanks – we have removed this.

- caption fig. 9: "black" is actually grey
  Thanks for spotting this. We have made this correction.

- Please elaborate what the control member in GLOFAS/IFS (ll. 129;467) and the "ensemble member" of the LSTM are (fig. 5)

Details on the LSTM ensemble are already given in methods in Section 4.3 (L245-250 in the submitted manuscript). Control members of GloFAS/IFS are simply unperturbed members of their respective ensembles, which we will clarify in the revised manuscript.

- The detailled description of and motivation behind the choice of training, testing and operational timespans could be placed elsewhere than in 4.2.1. Possibly best at beginning of chapter 4. Please also make it clear that LSTM and bias correction use the same time spans.
  Following this comment and a similar one by reviewer 2, we intend to provide a synthesis of the full methods/workflow for quick reference at the beginning of our revised section 4.

Potential changes for readability:

- 4.3. Include input features as table, not in text
  We agree that this will improve readability and will put the variables into a table (either in the methods section or an appendix).

- Present accuracies (aka skilfulness) of the three models as a table
  We will look at some different ways of representing these data in a table and include one in the revised manuscript if suitable.

---

## Author Response (AR1)

**Response to reviewers' comments for hess-2022-53: Hunt et al., Using a long short-term memory (LSTM) neural network to boost river streamflow forecasts over the western United States.**

We thank the editor and reviewers for reviewing out manuscript and for their helpful comments which we believe have greatly improved the studies contribution to the discipline. We respond to the comments of each of the reviewers below. Reviewer's comments are in black, and the authors' responses are in blue. All line numbers refer to the revised manuscript.

**RC1: 'Comment on hess-2022-53', Lennart Schmidt**

**General comments**

The manuscript delivers an interesting addition to the current surge of machine-learning in hydrological modelling by extending the application of LSTMs from pure streamflow modelling to actual forecasting. To do so, they ingest the output of physical forecasting systems as input to an LSTM. The main result, that LSTM outperforms the other approaches, is not surprising as the LSTM merely acts as a bias correction algorithm with many more degrees of freedom. Nevertheless, this is still a relevant finding that should be disseminated throughout the hydrological community. The manuscript is well-structured and comprehensible, particularly the introduction and methods parts are very comprehensive yet concise. Intuitive measures of model performance, a solid discussion of the error metrics as used in the study, a comprehensible discussion on the nature and choice of datasets as well as informative plots act as a solid foundation for the reader to follow along w.r.t to the methodological execution and its results.
We thank the reviewer for giving his time to provide a detailed review of our manuscript.

However, towards the end, the discussion and conclusion do miss out to put the work and the results into a broader perspective e.g., by contrasting it against ongoing machine-learning research, its limitations, or future directions for hybrid modelling (see detailed comments below).
We have expanded the discussion and the conclusion sections to highlight the limitations of the presented methods and to discuss possible future research. Specifically, please see replies to comments 1, 2, and 7.

**Scientific/Specific comments**

1. Please elaborate on the different types of "Hybrid" models/forecasts that are possible. In ML literature, there are current advances, termed "hybrid models", of including and solving differential equations inside the NN, promising the best of both worlds (high accuracy while keeping interpretability/robustness to out-of-distribution cases). These developments should be listed as future directions of research and the approach of this manuscript should be contrasted against these new developments in the introduction.
Rackauckas, Christopher, et al. "Universal differential equations for scientific machine learning." arXiv preprint arXiv:2001.04385 (2020).
Raissi, Maziar, Alireza Yazdani, and George Em Karniadakis. "Hidden fluid mechanics: Learning velocity and pressure fields from flow visualizations." Science 367.6481 (2020): 1026-1030. APA
We have added these discussion points and references to the revised introduction so that it also covers the limitations and future directions of hybrid modelling in hydrology (see lines 74-102). Thank you for the suggested references.

2. As the title suggests, the LSTM-approach is set up to "boost" the forecasts of IFS. As the LSTM receives Era5/IFS streamflow estimates, I personally would rather view it as a more sophisticated bias correction (more parameters, less constraints) than a separate modelling approach. I believe that the manuscript would benefit if this was put into perspective in the discussion part. Also, the possibility of using a simple statistical/ML model with less training effort, e.g., a simple linear model or Random Forest should at least be mentioned as an alternative.

Although the boundary is vague and ML/statistical correction, hybrid modelling, and pure ML models exist on a spectrum, the method implemented here would still work even if GloFAS were removed from the input (though perhaps with reduced skill). NWP variables are used as input, and so the LSTM is in some way replicating the hydrological processes, as in the Kratzert papers cited throughout. We have added a discussion on the breadth of the definition of hybrid forecasts to the introduction (see lines 74-88). Additionally, we have added a discussion of the use of simpler statistical methods and the role of the LSTM in the proposed method to the discussion. Please see lines 518-527.

3. Especially when taking into account that 7/10 catchments are known to be too small for the raw GloFAS to perform well, it is obvious that a statistical bias correction outperforms the raw forecast more the more degrees of freedom it is given. The "unfairness" of the comparison of raw GLOFAS vs. LSTM (world-wide simulation at 0.1° resolution vs. local model) should be highlighted in the introduction and discussion sections.

Yes – this is of course part of the motivation for trialling an LSTM in the first place. We have added this caveat to the revised introduction and discussions (see lines 119-121 and 596-600).

4. It would be beneficial to include and elaborate a bit further the motivation behind the two bias correction algorithms in the introduction to 4.2., i.e., quantile (remap values to reduce systematic bias) and spatial (inherent spatial bias in GLOFIS-ERA5(?)).

Quantile mapping is a fairly standard practice in hydrological bias correction (e.g. Thrasher *et al*., 2012, Li et al 2017). The spatial mapping provides an additional layer of bias correction, accounting for the fact that consistent spatiotemporal biases in hydrometeorological fields such as precipitation (should they exist), will result in consistent upstream/downstream biases in streamflow – information that can be used to improve forecasts. We have expanded the revised methods section to include this motivation (lines 244-248).

Thrasher, B., Maurer, E. P., McKellar, C., & Duffy, P. B. (2012). Bias correcting climate model simulated daily temperature extremes with quantile mapping. Hydrology and Earth System Sciences, 16(9), 3309-3314.
Li, W., Duan, Q., Miao, C., Ye, A., Gong, W., & Di, Z.: A review on statistical postprocessing methods for hydrometeorological ensemble forecasting. Wiley Interdisciplinary Reviews: Water, 4(December), e1246. 23 https://doi.org/10.1002/wat2.1246, 2017.

Also, the motivation for the two final steps should be explained and justified in greater detail. What do you mean by different climatologies, why split 3/4 vs. 1/4? Why do you shift the forecasts that have been quantile mapped-once more?

This is a fair question and has also been raised by reviewer 2. GloFAS-ERA5 and GloFAS forecasts have different climatologies because they take meteorological input from different sources (ERA5 and IFS forecasts respectively). ERA5 and IFS themselves have different climatologies because, although they share the same driving model, ERA5 is a reanalysis and is nudged towards observations, whereas IFS forecasts aren't. We realised at the beginning of the operational phase that the differences between GloFAS and GloFAS-ERA5 were leading to an overenthusiastic bias correction, and so damped it using the weights given. This point was also raised by reviewer 2, and we have expanded the revised methods section to include this explanation (296-304).

Also the fact that the bias correction has been newly developed (mentioned in conclusions) should be placed in the respective chapter in the methods part.

Yes – we have mentioned this in the revised methods section (line 246).

5. You argue that you used reanalysis data during train + test to make the results reproducible for potential users. But are the operational forecasts using IFS still reproducible? If not, it would be beneficial to provide the respective data on zenodo or a similar platform.
The IFS data used for these forecasts is freely available from the MARS web service hosted by ECMWF. Links are provided throughout the revised manuscript. The full streamflow forecasts themselves are available on the project GitHub page.

6. Generally, the manuscript misses to give detailed information on the training process. I would advise to include loss curve(s) (loss vs. epoch) of test and train. This is the common way to present information to see whether training was successful. Also, train and test error metrics should be provided to give an intuition whether under- or overfitting might have happened. The same applies to the bias correction, here the reader is not provided with any information on the optimization procedure or performance, even though there is a risk of overfitting. Similarly, information on the loss function, training hyperparameters (dropout, decay, learning rate, recurrent activation etc.) should be listed in appendix or refer to repository. To this end, the training process could have been performed more thoroughly: Hyperparameter-Tuning usually involves searching over model/training parameters as well as model configurations (n hidden layers, nodes etc.), not only epochs. The latter is usually less relevant. Ideally, hyperparameter tuning would be executed in a cross-validation set-up. To that respect, please elaborate what "tuned using sensitivity tests" refers to (l.240).
We agree that we have provided too little detail on the LSTM training process – although we would like to note that as the code is open access, a sufficiently interested reader could find the finer details there. However, in response to this suggestion and in keeping with ML literature conventions, we have expanded the methods section to include more information on (a) the final choice of hyperparameters, (b) how we arrived at these choices, and (c) how we avoided overfitting. Please see Section 4.4.
With the bias correction, which is ultimately a linear – if somewhat multivariate – model, we can be fairly sure that overfitting has not occurred for two reasons. Firstly, the number of coefficients sought is several orders of magnitude less than the number of training data points. Secondly, the validation – for which we used data set aside from the training/fitting process – suggested a very good fit. We have added these points to the revised methodology (lines 279-282).

7. The conclusion is (too) detailed on the technical side (n of skilful vs. non-skilful results, KGE vs. NSE) but misses to provide the most relevant point in 2-3 comprehensive sentences: Where exactly does LSTM perform best/worst (seasonal, diurnal, altitude etc.), Where are its limitations? Both the discussion and conclusion miss to put the results into a broader perspective: How could the LSTM be improved other than switching to Convolutional LSTM-layers? What are future research directions? Maybe some points to consider here:
We have added more detailed discussion on limitations and future research areas for ML (Section 6) as well as a synthesis on overall performance (Table 6 and lines 607-612).
We respond to the suggestions individually below.

- Limitations of LSTMs: Some limitations are fairly well-acknowledged by now (parallelization, long-range dependencies) so that LSTMs are not the go-to model for sequential data anymore. Thus, outlook on new developments like GRUs and, most importantly attention-based models (e.g. transformers) should be included
  Although LSTMs do have some limitations, they have been shown to be an incredibly powerful tool when it comes to modelling hydrological systems (e.g. Gauch *et al*, 2021). Of course, as ML models become increasingly sophisticated, there is little doubt that novel architectures/technique will perform such tasks much better, but there is very little (if any) published research on that yet. We have added a short discussion of these new developments in general sequential modelling to the revised discussion (lines 532-527).

Gauch, M., Kratzert, F., Klotz, D., Nearing, G., Lin, J., & Hochreiter, S. (2021). Rainfall–runoff prediction at multiple timescales with a single Long Short-Term Memory network. Hydrology and Earth System Sciences, 25(4), 2045-2062.

- Limitations of ML generally: Generalisation when provided with out-of-distribution data, e.g. due to system changes (climate change), lack of interpretability

  These are indeed often cited as shortcomings of ML. However, as shown in Kratzert *et al*. (2019), LSTMs trained across multiple basins actually perform well on previously unseen (even ungauged) basins that might typically be considered out-of-distribution. Similarly, such LSTMs can capture extreme values of streamflow driven by out-of-distribution extreme precipitation events (Frame *et al*., 2021). Taken together, these results suggest that such LSTMs are capable of capturing the underlying hydrological relationships that connect precipitation, runoff, and streamflow. Since climate change doesn't affect these physical relationships (only the magnitudes of the inputs), sufficiently advanced LSTMs should be largely immune. That said, other types of changes, e.g. increasing urbanisation, can affect the underlying relationships, and would degrade the skill of the LSTM – though this would of course happen in any other type of hydrological model if not updated. Please see lines 529-537.

  Although recent work has shown potential for interpretability in streamflow LSTMs (e.g. using attention theory; Li *et al*., 2021), we appreciate this is still generally a weakness compared to physics-based models – although in not having to rely on prescribed relationships, ML products can potentially learn new ones. We have added this to the discussion section (lines 537-539). However, this is not a limitation only applying to ML, but also to so called physically based models in particular when used in a climate change context

  Kratzert, F., Klotz, D., Herrnegger, M., Sampson, A. K., Hochreiter, S., & Nearing, G. S. (2019). Toward improved predictions in ungauged basins: Exploiting the power of machine learning. Water Resources Research, 55(12), 11344-11354.

  Frame, J., Kratzert, F., Klotz, D., Gauch, M., Shelev, G., Gilon, O., ... & Nearing, G. S. (2021). Deep learning rainfall-runoff predictions of extreme events. Hydrology and Earth System Sciences Discussions, 1-20.

  Li, W., Kiaghadi, A., & Dawson, C. (2021). High temporal resolution rainfall–runoff modeling using long-short-term-memory (LSTM) networks. Neural Computing and Applications, 33(4), 1261-1278.

- Greater picture:
    - Comparison to other hybrid modelling approaches in ML (see above)

      We have added a comparison to other hybrid modelling approaches in both the ML and hydrological disciplines. Please see lines 540-549.
    - How far away are we from entirely ML-based forecasts, given considerable advances in ML-based climatological forecasts? Should science still focus on improving relatively coarse physical model like ERA5 or rather explore ML-based bias correction at a large spatial scale?

      It is natural to speculate on this, but one may as well ask "why should we continue improving cars when we have planes?" Ultimately, physics-based and ML-based hydrological models should both continue to be improved since, although they can be used for the same purpose (e.g. forecasting, as in this paper), they have different strengths and weaknesses – as discussed above –and will thus continue to be suitable for different applications. We have added a discussion of the various ways and contributions of both physical methods and machine learning methods (lines 545-549).

- Reflect on the risk of overfitting in bias correction + lstms, including the fact that, ideally, one would have to estimate the bias of IFS w.r.t to ERA5.
  Yes – this was one of our biggest concerns, as we stated in the original discussion: "Similarly, the LSTM was trained on ERA5 data, but then ingested IFS output when run operationally. Although the two products will share some biases, they will inevitably be larger in IFS (the forecast) than ERA5 (the reanalysis), resulting in errors that propagate non-linearly through the LSTM. Originally, we chose to train the model on ERA5 so that our methods were reproducible, but there is no reason other forecasters should be bound by this desire. The optimal strategy is to train the LSTM on IFS hindcasts – though this would require careful adjustment of the architecture to account for different lead times. Similarly, such an approach must be careful of changing hindcast model versions." Please see lines 500-505.

**Technical Corrections**

- The abstract could be shortened
  We appreciate that at ~400 words, the abstract is a little on the longer side. However, the paper covers quite a wide variety of work and new methodology, so it isn't necessarily obvious how it could be shortened without loss of impact. However, we have reduced the abstract very slightly.

- l. 85, add bracket
  Thanks – we have fixed this.
- l. 110: "interesting challenge" is a subjective statement, rephrasing is advised
  We have removed "interesting" in the revised manuscript.
- l. 252 whether A extending
  We have removed the a.
- l.262 remove note
  We have removed this.
- l. 281: Is it 6 or 7 catchments that are skilful? 5.1.1. lists 6/10 that are skilful, here it is "still" 7/10?
  Good spot – we have edited this in the revised manuscript so that it now says (line 373): "Following bias-correction, GloFAS-ERA5 is now skilful at seven stations and highly skilful at four."
- l. 312 remove (CHECK)
  Thanks – we have removed this.
- caption fig. 9: "black" is actually grey
  Thanks for spotting this. We have made this correction.
- Please elaborate what the control member in GLOFAS/IFS (ll. 129;467) and the "ensemble member" of the LSTM are (fig. 5)
  Details on the LSTM ensemble are already given in methods in Section 4.3 (lines 245-250 in the submitted manuscript). We have added a more explicit description of the control members. Please see lines 163 and 186.
- The detailed description of and motivation behind the choice of training, testing and operational timespans could be placed elsewhere than in 4.2.1. Possibly best at beginning of chapter 4. Please also make it clear that LSTM and bias correction use the same time spans.
  Following this comment and a similar one by reviewer 2, we have provided a synthesis of the full methods/workflow for quick reference at the beginning of Section 4.

**Potential changes for readability:**

- 4.3. Include input features as table, not in text
  We have put the variables into a table (Table 5) in the methods section.

- Present accuracies (aka skilfulness) of the three models as a table
  We have included a table (Table 6) of the scores for the different forecasts and catchments in the results section.

**RC2: 'Comment on hess-2022-53', Frederik Kratzert**

**General points:**
- Data and model setup: I struggled a bit to follow the exact model setup and which data, in which temporal frequency, is used in the LSTM. Some of the information is already there, but distributed across the manuscript. I think it could increase the readability to have a dedicated paragraph (e.g. in Sect. 4.3, where some of the info is already available) that simply states: "We use A, B, …, C as model input at a temporal resolution of D. The model is trained to predict E days (hours?!) of streamflow forecast, using F features during the forecast period.". Again, some of the information is already there, but distributed across different sections (e.g. Data and Model sections). I found myself multiple times searching back-and-forth in the manuscript, to try to understand exactly what you do. E.g. in line 155 you say that streamflow was available in 3-hourly resolution, L. 248f states that all (gridded) data was averaged over the catchment "at a six-hourly frequency". From looking at the table on page 11 and L 249 of the manuscript, I can see that your input data frequency was 6-hourly, but the same table also says that you have a single model output. How exactly are you generating the forecasts for all *n* forecast timesteps?
We agree that the manuscript could really benefit from a summary of the overall setup and operational deployment, which we have included, as suggested, in the introduction. We have also acted on similar comments by the other two reviewers to clarify our methods throughout, for example including a table of input hydrometeorological variables (Table 5).

- As one of the authors of some of the LSTM literature that you cite, I feel like I need to comment on your general training setup. It seems like your setup was mainly guided by our first LSTM paper (2018), rather than any of the more recent papers (which you cite as well). You are probably aware that many things have changed since then and that your modeling setup does not really represent the best practices when working with LSTMs. For example, many studies (not only our papers) have shown that LSTMs really excel, when being trained on data from multiple basins at once and not on a gauge-by-gauge level as done in this manuscript. Even if the modeler is only interested in one (or a few) basin(s), it is better (with respect to the performance in the basin(s) of interest) to train on a larger set of basins. I think if you want to avoid running additional experiments, it would make sense to include at least a discussion on this topic.
We agree with this comment. When I (KMRH) set up the LSTM originally a few years ago, the methodology followed your 2018 paper (which was then more-or-less the only practice, let alone the best practice). Of course, we do appreciate that accepted best practice has changed in the

meantime and have discussed this in more detail in the revised introduction (lines 58-60) and discussion sections (lines 529-530).

- Similarly, and because you mention the computational expenses of training LSTMS, a comment on your model architecture: I know I used stacked LSTMs myself in the past (in the 2018 paper) but I was young and inexperienced. Since then, I spent many hours comparing different architectures and I never found a setup where it was worth using more than a single LSTM layer (which is what we have done since then in all of our publications). This is not critical and I don't want you to rerun experiments with different architectures, but it might be worth thinking about that when commenting on the computational expense. My experience from multiple large-sample model intercomparison studies is that the LSTM is by far the fastest model to train, compared to optimizing any hydrology model on scale. And the speed increases drastically if you use a single LSTM layer vs 3-stacked layers.

This is a very good point; one which we have discussed in the revised manuscript (lines 57-58 and lines 525-527).

**Line-by-line comments:**
- L. 43 "Schmidhuber et al. 1997" missing in the Reference section and "Hochreiter and Schmidhuber 1997" is wrongly formatted in the Reference section. Usually only "Hochreiter and Schmidhuber 1997" and "Gers et al. 2000" are cited as a reference for LSTMs.

Thanks – we have made these changes.

- L 48 "Chung et al. 2014" No need to cite this paper. This problem was the reason the LSTM was invented and is already discussed in "Hochreiter and Schmidhuber 1997", which would be a more appropriate reference here.

OK – we have made this substitution.

- L 57 "..even if the models were trained on multiple basins…" This is linked to my comment above. It is not "*even*" but "*because*" the models were trained on multiple basins. There are multiple studies (some of which you already cite) that show that the LSTM is able to transfer (learned) knowledge across basins. From my experience, LSTMs trained on a per-basin level are often actually not that much different from hydrology models. Another related reference that shows how model performance of LSTMs increase with the number of basins is the "The proper care and feeding of CAMELS: How limited training data affects streamflow prediction" by Gauch, Mai and Lin (2021).

We have included these changes in our expanded discussion of the literature (lines 58-60) see also response to major point 3). Thank you for the reference.

- L 59 "Extending this work, Gauch et al. (2021)..." Not really sure if I understand this sentence correctly. In Gauch et al. (2021), we extended the work from the other papers to models that are able to predict streamflow (or anything actually) on any given temporal resolution (and at multiple resolutions at the same time). What do you mean with "used a reanalysis framework to demonstrate the predictive power of LSTMs in streamflow modeling"?

We added this to demonstrate that work had been done on predicting streamflow using reanalyses, and not just observations (i.e. work deriving from the CAMELS datasets). The summary suggested in this comment is helpful though, so we have included it in our revision (lines 55-57).

- L 149 "as close as observation as possible" -> "as close to observations as possible".

But is this statement even true? Does GloFAS produce simulations that are "as close as possible" to observations? You mean as close as possible "for GloFAS" (i.e. GloFAS can't produce any better simulations) or as close as possible as "any model" can get? I would question the latter.

We definitely mean "for GloFAS" here and have removed this statement to avoid confusion.

- L 174 I shrugged when I read this passage. Personally, I really have problems to see that anything better than the long term mean is "skilful" and this is also not what Knoben et al (2019) said. Knoben et al. (2019) say that this threshold is often used to differentiate between "good" and "bad" models but they follow this with an explanation why such a global threshold is problematic in e.g. different kinds of flow regimes. Maybe you can at least extend the discussion around these thresholds a bit.

Yes, of course any definition of "skilful" is typically quite arbitrary. We certainly agree that just doing better than the climatology/mean doesn't really indicate meaningful skill, a definition that has irritated me in seasonal model evaluations for years. So perhaps this usage is a case of Stockholm Syndrome. We have discussed the choice of benchmarks in more detail - including correcting the statement attributed to Knoben et al (2019) – in the revised methods section. Please see lines 223-227.

- L 204ff Great explanation!
  Thank you!

- L 219 Eq. (7), Can you provide some explanation why you used a combination of NSE and KGE as your optimization function.

This is a good question, for which we have included an answer in the revised manuscript. Essentially, using NSE alone leaves the optimization procedure vulnerable to incorrect local minima (e.g. incorrect mean). However, we also found that using KGE alone tended to result in a bias correction that weighted correct mean and variance too highly compared to correlation (not useful for improving forecasts). We found that combining the two improved the weighting more in favour of correlation, while avoiding spurious local minima. Please see lines 275-279.

- L 226 Eq(8) Not entirely sure if I understand what I see. Is it correct that this matrix suggests that rather than including neighboring pixels (i.e. those in direct proximity to the gauge of interest) it relies on some distant pixels in all directions of the gauge? Are some of these pixels "downstream" of the gauge?

Yes – this interpretation is correct. All pixels are in the close neighbourhood of the gauge, but some are indeed downstream. This might seem a bit of a strange choice, but there are two advantages: (1) it is quick to set up, no additional topology data or preprocessing is required; and (2) we think (although we have not tested it) that it could help reduce temporal biases in some cases, for example, where the model puts the flow "too early", downstream information can actually be useful in correcting the bias. Where downstream information is not useful – as in the example given for NFSW4 in the paper – the weights collapse to zero (or very close) anyway.

- L 235 (equation) Is this based on (personal) experience or was this specifically tested for this study? I think it could be helpful to explain this with one sentence.

This was specifically tested at the beginning of the operational phase when we realised that the differences between GloFAS and GloFAS-ERA5 were leading to an overenthusiastic bias correction. We have mentioned this in the revised methods section (lines 306-310).

- Figure 2: Great Figure and interesting results!
  Thank you!

- L 262 "Note: snowc not available in IFS output" Is this an artifact from the manuscript preparation or otherwise can you extend what you want to say here?
This was left over from the original manuscript preparation and has been removed in the revision.

- Sect 5.1.X "Evaluation of the test period": I might be missing something but can you explain what exactly is shown as a simulated hydrograph? The model is built to provide a 10-day forecast right? So for every calendar day, you would have multiple forecasted values. Which value do you pick for these plots?
All models are driven with ERA5 during the testing period, since they are not being run in forecast mode and we simply want to test their respective abilities to replicate observed streamflow. This contrasts with Figs 6/7/8, where the models *are* run in forecast mode (driven by IFS output), and for which we plot hydrographs at three different lead times.

- Fig 6, 7, 8 are a bit hard to read because of the small figure size and the many overlapping lines.
We appreciate that these figures are quite compact – but they are vector images so readers can get finer detail by zooming in.

- L 312 I think you want to remove the "(CHECK)".
Thank you for spotting this – we have removed this.

- L 420 "training them [LSTMs] is computationally very expensive". My understanding/experience from participating in various large-sample benchmarking studies is that training LSTMs is much faster than optimizing hydrology models on a similar scale. This is probably related to your specific LSTM architecture (3-stacked layer), which by itself might not be necessary.
We have included this caveat in our revised discussion. Please see lines 538-540.

**RC3: 'Comment on hess-2022-53', Anonymous Referee #3**

**General Comments**

This paper is an exciting work moving machine learning models from a research lab to an operational setting. The manuscript is quite comprehensive and compares extensively to existing operational methods, highlighting advantages and challenges associated with the machine learning model (LSTM). Overall LSTM performs better than the other methods, which is expected given the nature of the machine learning model (if we have enough data).

We thank the reviewer for their positive assessment of our manuscript, and for taking the time to evaluate it critically.

**Specific comments**

The manuscript claims that it has been the first time that LSTM has been used in a hybrid system to create a medium-range weather forecast. In sync with the comment - RC1 Clear distinction should be made with hybrid models. This can be achieved by infographics (pictorial representation) of different models and variables used with a bounding box illustrating what is called a hybrid system and how it varies for different kinds of models. This would also help the readers to understand the entire workflow.

We have added a figure describing the workflow (Figure 2). However, we don't think that a figure more broadly reviewing types of hybrid forecasting systems is either necessary or in scope. We have added a discussion on the breadth of hybrid systems (lines 74-102).

Concerning line 53 "In this regard, studies fall into two categories – either seeking to create a model capable of replicating existing streamflow observations or seeking to create a model capable of forecasting streamflow at some future time. Several highly illustrative studies approach the former topic….." Though the manuscript proposes two different categories but essentially, from a machine learning perspective, they might not be very different. Replication is also a form of prediction for a machine learning model. Distinction based on this might not be appropriate here. A rather significant difference is the use of streamflow at previous timesteps. Or in general, the first approach could be using only the drivers (precipitation, temperature radiation, wind) where we have almost no explicit information about the inherent state of the catchment (things like how moist is the soil, how much snow we have in the catchment, which might melt) to make predictions of streamflow

(series of papers published by researchers at Johannes Kepler University Linz is in this direction). While the second category could be where we explicitly include the inherent state information (flow at previous time steps) about the catchment, which will have more potential for better predictions. We agree with the reviewer that there is very little, if any, distinction between replication and prediction (i.e. forecasting) in the research context of machine learning models. However, these are clearly different tasks when it comes to application because they solve different problems and (as the reviewer highlights) ingest different data. Therefore, we believe the distinction is very much appropriate to make here, and, given the subject of the paper is the operational application of an LSTM, it makes sense to include a discussion of previous work that delineates between earlier "theoretical" (i.e. replication) and "applied" (i.e. forecasting/prediction) experiments. We have added further discussion on the motivation for this distinction.

That said, the distinction between modelling basins that have, as the reviewer says, "almost no explicit information" and basins that have "inherent information" – i.e. essentially ungauged and gauged basins respectively – is interesting from a research point of view and we have includes a discussion on this in the revised introduction (lines 51-52).

For the LSTM model, 23 variables have been chosen for predictions; while not doing any extensive hyperparameter search (optimum number of variables), some rationale needs to be provided on why those variables were chosen (if any kind of qualitative selection was made).

Following Kratzert et al (2018) section 2.4, we included all surface or near-surface variables available in ERA5 that were potentially relevant to streamflow prediction. We have clarified this in the revised manuscript (Table 5 and Section 4.4.2).

Figure 2 of the paper is really interesting, and we can see that for some catchments, through different epochs NSE (and RMSE) changes a lot for models initialised with different weights. Is it normal for all machine learning models to vary a lot after hyperparameter tuning?

Thank you. Yes – this has been an active field of study for some time (e.g. Kolen and Pollack 1990). There is also evidence that this is the case for RNNs (e.g. Graves et al., 2013).

Kolen, J., & Pollack, J. (1990). Back propagation is sensitive to initial conditions. Advances in neural information processing systems, 3.

Graves, A., Mohamed, A. R., & Hinton, G. (2013, May). Speech recognition with deep recurrent neural networks. In 2013 IEEE international conference on acoustics, speech and signal processing (pp. 6645-6649). IEEE.

Secondly, we also see that the variability in NSE (and RMSE) is also high when models are trained for 100 epochs, and they perform worse than the best models trained with 10 epochs. This could also result from overfitting. While the figure provides a nice way to represent the uncertainties associated with the model, it might also make them look more uncertain than they actually are. As there are methods in machine learning to decrease this variability, it would be interesting to see how models perform if trained for 100 epochs with early stopping criteria.

The reviewer is correct. Following this comment and several on similar lines from reviewer 1 (Lennart Schmidt), we have improved the training discussion in the revised manuscript, including a new figure on loss by epoch (Figure 3 and Section 4.4.1).

The discussion section is focused on the use of the convolution layer, a big challenge would be handling the different sizes of catchments. Generalised 1D representation of 2-D values might be a research direction. The other could be the use of graph neural network. A good example could be in the direction of  the paper "Spatial and Temporal Aware Graph Convolutional Network for Flood Forecasting"

Thanks for the suggestion and reference – we have added these points to the revised discussion (lines 516-517).
Feng, Z. Wang, Y. Wu and Y. Xi, "Spatial and Temporal Aware Graph Convolutional Network for Flood Forecasting," 2021 International Joint Conference on Neural Networks (IJCNN), 2021, pp. 1-8, doi: 10.1109/IJCNN52387.2021.9533694.

**Minor technical comment**
Figure 2 A suggestion would be to either create a plot with a colour gradient for the density of points, else making the marker size smaller can help in illustrating where most of the points are (reducing the overlap).
Thank you, we have implemented this suggestion in Figure 4.

---

## Referee Report (RR1)

Generally, the authors have addressed all my comments in an adequate manner. They greatly improved the introduction, doing a great job of contrasting the paper against the nature of hybrid modelling - which, as they correctly point out, is not well defined. While I would challenge their comment that "LSTM is in some way replicating the hydrological processes, as in the Kratzert papers cited throughout" as these papers did not include any streamflow forecasts as inputs (at least to my knowledge), I would leave further debate on this and the nature of hybrid models in hydrology to the open scientific community. Also, the methods section has improved a lot, making the all steps of the experiments a lot clearer. The discussion has also been extended, now sufficiently contrasting the work against current research. However, while the conclusion summarizes the key methods and results in a precise and short manner, it ends rather abruptly without providing 2-3 sentences to put the results into the broader context as outlined in the introduction (streamflow forecasting using ML, complementing the current surge of ML in hydrological sciences, one of multiple possible applications of hybrid modelling or similar). I would suggest the authors to smoothen this out in the final manuscript.

Below, some formal or minor comments:

- l.84 are DiscusseS above
- l. 85: is the abbreviation NWP introduced before?
- l. 89: Boucher et al - Year missing
- l. 96: "by sequentially"?
- ll. 99-101: Please revise citation formats
- Fig. 2 greatly facilitates understanding what was done, thank you for adding it. For completeness, one could differentiate between the Training/Testing and operational period here, i.e. indicating that it is either ERA5 or IFS that is used as input
- l. 349 - Training period was previously noted to be 1990-2019
- ll. 356-67: The reference to "early stopping" is not quite clear to me. How would that make models less sensitive to inital weights?
- ll. 620-623 - redundant with figure caption. For me, this does not have to be repeated here.
- ll. 620-625: While valuable information, this paragraph does not use it to place the results into broader perspective. I suggest concluding this paragraph by an extended version of l 619, quickly getting back to the broader context and contribution of this work to it.
- Table 6: Caption - Aren`t better performing gauges placed towards the right?
- References: Random find: Frame 2022 is listed twice

---

## Author Response (AR2)

**Response to reviewers' reports for hess-2022-53: Hunt et al., Using a long short-term memory (LSTM) neural network to boost river streamflow forecasts over the western United States.**

We are grateful to the reviewers for giving their time to review our revised manuscript and for again providing detailed feedback to improve the quality of the manuscript. We respond to the comments of both reviewers in turn below. Reviewer's comments are in black and the authors' responses are in blue.

**Report #1: Frederik Kratzert**

Given the current state of the manuscript, I think the paper is acceptable for publications. Congratulations to the author team.

Thank you for your comments throughout the review process which have been invaluable.

Two nit-picks (line numbers are referring to line numbers in the track changes manuscript):

L: 324 "...giving an input vector of length 28…" I think "input sequence" instead of "input vector" is a better fit. You are talking about a sequence of input vectors here (shape [sequence length, input features]), which is a matrix/tensor but not a vector.

We have made this substitution on line 324, and on lines # and # to maintain consistency.

L: 341 "tht" -> "that"

We have corrected this error.

**Report #2: Lennart Schmidt**

Generally, the authors have addressed all my comments in an adequate manner. They greatly improved the introduction, doing a great job of contrasting the paper against the nature of hybrid modelling - which, as they correctly point out, is not well defined. While I would challenge their comment that "LSTM is in some way replicating the hydrological processes, as in the Kratzert papers cited throughout" as these papers did not include any streamflow forecasts as inputs (at least to my knowledge), I would leave further debate on this and the nature of hybrid models in hydrology to the open scientific community. Also, the methods section has improved a lot, making all the steps of the experiments a lot clearer. The discussion has also been extended, now sufficiently contrasting the work against current research.

Thank you for your helpful comments throughout the review process.

However, while the conclusion summarizes the key methods and results in a precise and short manner, it ends rather abruptly without providing 2-3 sentences to put the results into the broader context as outlined in the introduction (streamflow forecasting using ML, complementing the current surge of ML in hydrological sciences, one of multiple possible applications of hybrid modelling or similar). I would suggest the authors to smoothen this out in the final manuscript.

We have edited the final paragraph in the conclusions to put the work into greater context and potential future implications of the work. Please see the responses to the comments below regarding lines 619-625 for more details.

Below, some formal or minor comments:

l.84 are DiscusseS above

We have corrected this error.

l. 85: is the abbreviation NWP introduced before?

It was not. We have expanded the abbreviation.

l. 89: Boucher et al - Year missing

Thank you we have corrected this reference.

l. 96: "by sequentially"?

This has been corrected to "This can either be sequentially …"

ll. 99-101: Please revise citation formats

We have changed the citations within this sentence to improve readability.

Fig. 2 greatly facilitates understanding what was done, thank you for adding it. For completeness, one could differentiate between the Training/Testing and operational period here, i.e. indicating that it is either ERA5 or IFS that is used as input

Thank you. We have added the difference between input data for the training/testing and operational periods.

l. 349 - Training period was previously noted to be 1990-2019

Thank you. We have corrected this discrepancy.

ll. 356-67: The reference to "early stopping" is not quite clear to me. How would that make models less sensitive to initial weights?

We have replaced this with "The sensitivity… to initial weights… could be reduced in future work by using regularisation techniques such as weight decay"

ll. 620-623 - redundant with figure caption. For me, this does not have to be repeated here.

We have removed the repetition of the figure caption.

ll. 620-625: While valuable information, this paragraph does not use it to place the results into broader perspective. I suggest concluding this paragraph by an extended version of l 619, quickly getting back to the broader context and contribution of this work to it.

Thank you. We have combined the paragraphs to put the work in broader context.

Table 6: Caption - Aren`t better performing gauges placed towards the right?

Yes. Thank you this has been corrected.

References: Random find: Frame 2022 is listed twice

Thank you. We have removed one of these entries, however, this is only visible in the new manuscript and not the tracked changes document.